# Effectiveness of rapid SARS-CoV-2 genome sequencing in supporting infection control for hospital-onset COVID-19 infection: Multicentre, prospective study

Oliver Stirrup[1], James Blackstone[2], Fiona Mapp[1], Alyson MacNeil[2], Monica Panca[2], Alison Holmes[3], Nicholas Machin[4], Gee Yen Shin[5], Tabitha Mahungu[6], Kordo Saeed[7], Tranprit Saluja[8], Yusri Taha[9], Nikunj Mahida[10], Cassie Pope[11], Anu Chawla[12], Maria-Teresa Cutino-Moguel[13], Asif Tamuri[14], Rachel Williams[15], Alistair Darby[16], David L Robertson[17], Flavia Flaviani[18], Eleni Nastouli[5], Samuel Robson[19], Darren Smith[20], Matthew Loose[21], Kenneth Laing[22], Irene Monahan[22], Beatrix Kele[13], Sam Haldenby[16], Ryan George[4], Matthew Bashton[23], Adam A Witney[22], Matthew Byott[5], Francesc Coll[24], Michael Chapman[25], Sharon J Peacock[26], COG-UK HOCI Investigators[†], The COVID-19 Genomics UK (COG-UK) consortium[‡], Joseph Hughes[17], Gaia Nebbia[18], David G Partridge[27], Matthew Parker[28], James Richard Price[3], Christine Peters[29], Sunando Roy[30], Luke B Snell[18], Thushan I de Silva[31], Emma Thomson[17], Paul Flowers[32], Andrew Copas[1], Judith Breuer[30]*

*For correspondence:
j.breuer@ucl.ac.uk

†Full list of investigators and their affiliations can be found in Supplementary file 1

‡Full list of consortium members and their affiliations can be found in Supplementary file 2

[1]Institute for Global Health, University College London, London, United Kingdom; [2]The Comprehensive Clinical Trials Unit, University College London, London, United Kingdom; [3]Imperial College Healthcare NHS Trust, London, United Kingdom; [4]Manchester University NHS Foundation Trust, Manchester, United Kingdom; [5]University College London Hospitals NHS Foundation Trust, London, United Kingdom; [6]Royal Free London NHS Foundation Trust, London, United Kingdom; [7]University Hospital Southampton NHS Foundation Trust, Southampton, United Kingdom; [8]Sandwell & West Birmingham Hospitals NHS Trust, Birmingham, United Kingdom; [9]Department of Virology and Infectious Diseases, Newcastle-upon-Tyne Hospitals NHS Foundation Trust, Newcastle, United Kingdom; [10]Nottingham University Hospitals NHS Trust, Nottingham, United Kingdom; [11]St George's University Hospitals NHS Foundation Trust, London, United Kingdom; [12]Liverpool University Hospitals NHS Foundation Trust, Liverpool, United Kingdom; [13]Barts Health NHS Trust, London, United Kingdom; [14]Research Computing, University College London, London, United Kingdom; [15]Department of Genetics and Genomic Medicine, UCL Great Ormond Street Institute of Child Health, University College London, London, United Kingdom; [16]Centre for Genomic Research, University of Liverpool, Liverpool, United Kingdom; [17]MRC-University of Glasgow Centre For Virus Research, University of Glasgow, Glasgow, United Kingdom; [18]Guy's and St Thomas' Hospital NHS Foundation Trust, London, United Kingdom; [19]Centre for Enzyme Innovation and School of Pharmacy and Biomedical Science, University of Portsmouth, Portsmouth, United Kingdom; [20]Department of Applied Sciences, Northumbria University, Newcastle-upon-Tyne, United Kingdom; [21]School of Life

Sciences, University of Nottingham, Nottingham, United Kingdom; [22]Institute for Infection and Immunity, St George's University of London, London, United Kingdom; [23]The Hub for Biotechnology in the Built Environment, Department of Applied Sciences, Northumbria University, Newcastle, United Kingdom; [24]Department of Infection Biology, Faculty of Infectious and Tropical Diseases, London School of Hygiene & Tropical Medicine, London, United Kingdom; [25]Health Data Research UK Cambridge Hub, Cambridge, United Kingdom; [26]Department of Medicine, University of Cambridge, Cambridge, United Kingdom; [27]Sheffield Teaching Hospitals NHS Foundation Trust, Sheffield, United Kingdom; [28]Sheffield Bioinformatics Core, University of Sheffield, Sheffield, United Kingdom; [29]NHS Greater Glasgow and Clyde, Glasgow, United Kingdom; [30]Department of Infection, Immunity and Inflammation, UCL Great Ormond Street Institute of Child Health, University College London, London, United Kingdom; [31]Department of Infection, Immunity and Cardiovascular Disease, University of Sheffield, Sheffield, United Kingdom; [32]School of Psychological Sciences and Health, University of Strathclyde, Glasgow, United Kingdom

## Abstract

**Background:** Viral sequencing of SARS-CoV-2 has been used for outbreak investigation, but there is limited evidence supporting routine use for infection prevention and control (IPC) within hospital settings.

**Methods:** We conducted a prospective non-randomised trial of sequencing at 14 acute UK hospital trusts. Sites each had a 4-week baseline data collection period, followed by intervention periods comprising 8 weeks of 'rapid' (<48 hr) and 4 weeks of 'longer-turnaround' (5–10 days) sequencing using a sequence reporting tool (SRT). Data were collected on all hospital-onset COVID-19 infections (HOCIs; detected ≥48 hr from admission). The impact of the sequencing intervention on IPC knowledge and actions, and on the incidence of probable/definite hospital-acquired infections (HAIs), was evaluated.

**Results:** A total of 2170 HOCI cases were recorded from October 2020 to April 2021, corresponding to a period of extreme strain on the health service, with sequence reports returned for 650/1320 (49.2%) during intervention phases. We did not detect a statistically significant change in weekly incidence of HAIs in longer-turnaround (incidence rate ratio 1.60, 95% CI 0.85–3.01; p=0.14) or rapid (0.85, 0.48–1.50; p=0.54) intervention phases compared to baseline phase. However, IPC practice was changed in 7.8 and 7.4% of all HOCI cases in rapid and longer-turnaround phases, respectively, and 17.2 and 11.6% of cases where the report was returned. In a 'per-protocol' sensitivity analysis, there was an impact on IPC actions in 20.7% of HOCI cases when the SRT report was returned within 5 days. Capacity to respond effectively to insights from sequencing was breached in most sites by the volume of cases and limited resources.

**Conclusions:** While we did not demonstrate a direct impact of sequencing on the incidence of nosocomial transmission, our results suggest that sequencing can inform IPC response to HOCIs, particularly when returned within 5 days.

**Funding:** COG-UK is supported by funding from the Medical Research Council (MRC) part of UK Research & Innovation (UKRI), the National Institute of Health Research (NIHR) (grant code: MC_PC_19027), and Genome Research Limited, operating as the Wellcome Sanger Institute.

**Clinical trial number:** NCT04405934.

## Editor's evaluation

This article contains valuable information on the potential value of real-time genome sequencing to inform infection control practices. The study, unique in its size, addresses the implementation of this approach during the height of the COVID-19 pandemic. Naturally, the extreme situation limited the options for choices in infection control practices.

## Introduction

Viral sequencing has played an important role in developing our understanding of the emergence and evolution of the SARS-CoV-2 pandemic (*Oude Munnink et al., 2021*). Sequencing technologies can now be used for local outbreak investigation in near real time, and this was implemented by some research centres for evaluation of nosocomial transmission from the early stages of the pandemic (*Meredith et al., 2020*). It has been demonstrated that sequencing can provide additional information on outbreak characteristics and transmission routes in comparison to traditional epidemiological investigation alone (*Meredith et al., 2020*; *Lucey et al., 2021*; *Snell et al., 2022b*). However, limited data are available on the feasibility of routine use of sequencing for infection prevention and control (IPC), or on its direct impact on IPC actions and nosocomial transmission rates.

Throughout the pandemic, nosocomial transmission of SARS-CoV-2 has been a major concern (*Abbas et al., 2021*), with hospital-acquired infections (HAIs) accounting for more than 5% of lab-confirmed cases from March to August 2020 in the UK (*Bhattacharya et al., 2021*) and representing 11% of COVID-19 cases within hospitals in this period (*Read et al., 2021*). HAIs also frequently occur within a very vulnerable population with high levels of mortality (*Bhattacharya et al., 2021*; *Oliver, 2021b*; *Ponsford et al., 2021*). There is therefore an unmet need to develop interventions that can reduce the occurrence of nosocomial transmission. The aims of this study were to determine the effectiveness of SARS-CoV-2 sequencing in informing acute IPC actions and reducing the incidence of HAIs when used prospectively in routine practice, and to record the impact of sequencing reports on the actions of IPC teams.

When this study was planned in the summer of 2020, there was imperfect knowledge regarding the dominant mode of transmission of SARS-CoV-2 (*Greenhalgh et al., 2021*), and it was not possible to predict the future course of the pandemic. In conducting this study, substantial difficulties were encountered in implementing the intervention and in responding effectively to any insights generated. As such, this report serves as a record of the challenge of conducting research within a pandemic as well as being a conventional study summary report.

## Methods

We conducted a prospective multiphase non-randomised trial to evaluate the implementation and impact of SARS-CoV-2 sequencing for IPC within 14 acute NHS hospital groups in the UK. All sites were linked to a COG-UK sequencing hub, 13 were university hospitals and 1 a district general hospital. We implemented a bespoke sequence reporting tool (SRT) intervention, developed and previously evaluated for this study (*Stirrup et al., 2021*), and assessed the importance of turnaround time for sequencing and reporting. The study included integral health economic and qualitative process evaluation (*Flowers et al., 2021*).

The study design comprised a planned 4-week baseline data collection period, followed by intervention periods defined by the time from diagnostic sampling to return of sequence data to IPC teams, comprising 8 weeks of 'rapid' (<48 hr) turnaround sequencing and 4 weeks of 'longer' (5–10 days) turnaround sequencing for each site. Target turnaround time was 48 hr from diagnostic sampling to return of the SRT report during the 'rapid' sequencing phase, and 5–10 days for the 'longer-turnaround' phase. Eight sites implemented 'rapid' followed by 'longer-turnaround' phases with five doing the opposite. One site did not implement longer-turnaround sequencing because they considered it a reduction in their standard practice, comprising outbreak sequencing with weekly meetings to discuss phylogenetic analyses; they nonetheless completed the baseline phase of the study without use of the SRT or automated feedback to IPC teams on all hospital-onset COVID-19 infection (HOCI) cases. The order of the intervention phases was pragmatically determined in some sites by the need to first run the 'longer-turnaround' phase to develop sample transport and sequencing procedures before attempting the 'rapid' sequencing phase, and the ordering was decided in the remaining sites to ensure a mixture of intervention phases over calendar time – there was no randomisation process in deciding the order of study phases.

Data were recorded in all phases for all patients meeting the definition of a HOCI, that is, first confirmed test for SARS-CoV-2 >48 hr after admission and without suspicion of COVID-19 at the time of admission. During the intervention phases, and for at least 3 weeks prior to any intervention

period to enable linkage to recent cases, participating sites aimed to sequence all SARS-CoV-2 cases including both HOCI and non-HOCI cases.

The SRT aimed to integrate sequence and patient data to produce concise and immediately interpretable feedback about cases to IPC teams via a one-page report. Sites were also able to apply other methods (e.g. phylogenetics) to the sequence data generated, where this was part of their usual practice. Guidance regarding IPC actions was not specified as part of this study. Sites were expected to follow current national guidelines, which evolved throughout the course of the pandemic. Sequencing data from healthcare workers (HCWs) could be utilised in the SRT system, and this was implemented by 8/14 sites. Whether this was done depended on availability of HCW samples for each lab as staff testing was generally managed separately to patient testing. HCW testing protocols followed national guidelines.

Data collection on patient characteristics and on implementation and impact of the intervention was conducted using a central study-specific database. Ethical approval for the study was granted by NHS HRA (REC 20/EE/0118), and the study was prospectively registered (ClinicalTrials.gov Identifier: NCT04405934).

The primary outcomes of the study as defined in the protocol (*Blackstone et al., 2022*) were (1) incidence of IPC-defined SARS-CoV-2 HAIs per week per 100 currently admitted non-COVID-19 inpatients, and (2) for each HOCI, identification of linkage to individuals within an outbreak of SARS-CoV-2 nosocomial transmission using sequencing data as interpreted through the SRT that was not identified by pre-sequencing IPC evaluation during intervention phases. The second outcome used all observed HOCI cases as the denominator, and so represented the proportion of cases in which sequencing provided information regarding potential transmission routes where none had been previously uncovered.

Secondary outcomes were (1) incidence of IPC-defined SARS-CoV-2 hospital outbreaks per week per 1000 non-COVID-19 inpatients; (2) for each HOCI, any change to IPC actions following receipt of SRT report during intervention phases; and (3) any recommended change to IPC actions (regardless of whether changes were implemented). There was considered to be an impact on IPC actions if this was recorded for any of a number of predefined outcomes (e.g. enhanced cleaning, visitor and staffing restrictions, provision of personal protective equipment), or if it was stated that the report had effected any change to IPC practice on that ward or elsewhere within the hospital. The proportion of HOCI cases for which IPC reported the SRT report to be 'useful' was added as a further outcome.

To support standardisation across sites, 'IPC-defined SARS-CoV-2 HAIs' were considered to be all HOCIs with ≥8 days from admission to symptom onset (if known) or sample date (i.e. UK Health Security Agency definition of a probable/definite HAI; *Public Health England, 2020*).

An IPC-defined SARS-CoV-2 hospital outbreak was defined as at least two HOCI cases on the same ward, with at least one having ≥8 days from admission to symptom onset or sample date. Outbreak events were considered to be concluded once there was a period of 28 days prior to observation of another HOCI (*Public Health England, 2020*).

Further details of outcome definitions are given in Appendix 1.

## Statistical analysis

We used three approaches: intention-to-treat analysis to assess the overall impact of sequencing on IPC activity and the incidence of HAIs, per protocol site-based analysis on a subset of high-performance sites, and pooled analysis to describe how turnaround time was related to impact on IPC irrespective of study phase. Inclusion of sites in the per protocol analysis was based on the proportion of sequence reports returned and speed of return in the rapid phase. Thresholds to define this group were determined following review of the data but before analysis of outcomes.

Incidence outcomes were analysed using mixed effects negative binomial regression models, which in this context correspond to Poisson regression with an additional overdispersion parameter. Data for the first week of each intervention period, or in the first week of return to intervention following a break, were considered transition periods and not considered as direct evidence regarding the intervention effect. Analysis was conducted with calendar time divided into 'study weeks' running Monday–Sunday. Models were adjusted for calendar time, the proportion of current inpatients that were SARS-CoV-2 positive, as well as local community SARS-CoV-2 incidence for each study site, using 5 knot restricted cubic splines (*Kahan et al., 2016*). The number of inpatients not positive for

SARS-CoV-2 was considered an exposure variable (defining 'person-time' at risk of nosocomial infection). Differences between study phases were evaluated using adjusted incidence rate ratios.

The primary outcome of identification of SARS-CoV-2 nosocomial transmission using sequencing data and secondary outcomes relating to changes to IPC actions and the 'usefulness' of SRT reports were analysed using mixed effects logistic regression models, without covariable adjustment or removal of cases from the first week of each intervention phase. Marginal proportions from fitted models were reported for rapid- and longer-turnaround intervention phases, and differences in outcomes between these phases were evaluated. If the SRT report was not returned, this was interpreted as a 'failure', that is, no change to IPC action; however, we also present percentages for these outcomes restricted to HOCIs where the SRT report was returned.

**Table 1.** Demographic and baseline characteristics of the participants by study phase.

| Characteristic at screening | Study phase | | | |
| --- | --- | --- | --- | --- |
| | **Baseline** | **Longer-turnaround** | **Rapid** | **Total** |
| *N* HOCI cases | 850 | 373 | 947 | 2170 |
| *N* HOCI cases per site, median (range); *N* sites | 36 (1–207); 14 | 19 (0–86); 13 | 30.5 (4-297); 14 | 103.5 (40-451); 14 |
| *HAI classification, n (%)* | | | | |
| Indeterminate (3–7 days) | 362 (42.6) | 166 (44.5) | 371 (39.2) | 899 (41.4) |
| Probable (8–14 days) | 236 (27.8) | 121 (32.4) | 270 (28.5) | 627 (28.9) |
| Definite (>14 days) | 252 (29.6) | 86 (23.1) | 306 (32.3) | 644 (29.7) |
| Age (years), median (IQR, range) | 77.5 (65.4–85.6, 0.4–100.5) | 77.6 (64.6–86.7, 0.7–100.7) | 76.4 (62.6–85.5, 0.6–103.5) | 76.7 (64.4–85.6, 0.4–103.5) |
| Age ≥70 years, *n/N* (%) | 589/850 (69.3) | 240/373 (64.3) | 598/947 (63.1) | 1427/2170 (65.8) |
| Sex at birth: female, *n/N* (%) | 457/850 (53.8) | 177/372 (47.6) | 460/947 (48.6) | 1094/2169 (50.4) |
| *Ethnicity, n (%)* | | | | |
| White | 668 (78.6) | 275 (73.7) | 732 (77.3) | 1675 (77.2) |
| Mixed ethnicity | 9 (1.1) | 6 (1.6) | 8 (0.8) | 23 (1.1) |
| Asian | 46 (5.4) | 26 (7.0) | 34 (3.6) | 106 (4.9) |
| Black Caribbean or African | 36 (4.2) | 18 (4.8) | 46 (4.9) | 100 (4.6) |
| Other | 6 (0.7) | 1 (0.3) | 4 (0.4) | 11 (0.5) |
| Unknown | 85 (10.0) | 47 (12.6) | 123 (13.0) | 255 (11.8) |
| Symptomatic at time of sampling, *n/N* (%) | 167/739 (22.6) | 58/322 (18.0) | 106/659 (16.1) | 331/1720 (19.2) |
| Significant comorbidity present, *n/N* (%) | 650/776 (83.8) | 260/323 (80.5) | 574/757 (75.8) | 1484/1856 (80.0) |
| Pregnant, *n/N* (%) | 6/451 (1.3) | 1/177 (0.6) | 4/445 (0.9) | 11/1073 (1.0) |
| *Hospital admission route, n (%)* | | | | |
| Emergency department | 605 (71.2) | 258 (69.2) | 549 (58.0) | 1412 (65.1) |
| Hospital transfer | 59 (6.9) | 21 (5.6) | 51 (5.4) | 131 (6.0) |
| Care home | 3 (0.4) | 0 (0) | 0 (0) | 3 (0.1) |
| GP referral | 38 (4.5) | 15 (4.0) | 76 (8.0) | 129 (5.9) |
| Outpatient clinic ref. | 27 (3.2) | 20 (5.4) | 30 (3.2) | 77 (3.5) |
| Other | 42 (4.9) | 9 (2.4) | 48 (5.1) | 99 (4.6) |
| Unknown | 76 (8.9) | 50 (13.4) | 193 (20.4) | 319 (14.7) |

GP, general practitioner; HAI, hospital-acquired infection; HOCI, hospital-onset COVID-19 infection.

For both incidence and 'per HOCI' outcomes, we accounted for the structure of the data with hierarchical exchangeable normally distributed random effects for each study site, and for each study phase within each study site. Analyses were conducted using Stata V16, with figures generated using the *ggplot2* package for R V4.0.

## Results

A total of 2170 HOCIs were recorded for the study between 15 October 2020 and 26 April 2021. These cases had median age of 76.7 (interquartile range [IQR] 64.4–85.6) years, and 80% had at least one clinically significant comorbidity (*Table 1*).

All 14 sites completed baseline and rapid sequencing intervention phases (*Appendix 1—figure 1*). Thirteen sites completed the longer-turnaround sequencing intervention phase. 49.2% (650/1320) SRT reports for HOCIs were returned in the intervention phases, with only 9.3% (123/1320) returned within the target time frames (*Table 2*). This figure was greater in the longer-turnaround phase at 21.2% (79/373) than in the rapid phase (4.6%; 44/947). The median turnaround time from diagnostic sampling for reports returned was 5 days in the rapid phase and 13 days in the longer-turnaround phase, substantially longer than the targets of 48 hr and 5–10 days, respectively. A detailed break-down of reporting turnaround times is reported separately (*Colton et al., 2022*).

Ordering the sites by proportion of cases with sequencing results returned and median turnaround time during the rapid phase (*Figure 1*) identified no obvious clustering of highest vs. lowest performing sites. We therefore also carried out a 'per protocol' sensitivity analysis on the seven highest performing sites; these sites returned ≥40% of SRTs within a median time from diagnostic sample of ≤8 days within their rapid phase. The criteria for this analysis were decided after data collection but prior to data analysis, as per the statistical analysis plan (SAP). However, we acknowledge that the 'higher performing sites' did not meet the target turnaround time for reporting in the rapid phase; criteria were therefore set to split the sites into upper and lower 50% based on the level of implementation.

We did not detect a statistically significant change in weekly incidence of HAIs in the longer-turnaround (incidence rate ratio 1.60, 95% CI 0.85–3.01; p=0.14) or rapid (0.85, 0.48–1.50; 0.54) intervention phases in comparison to baseline phase across the 14 sites (*Table 3*), and incidence rate ratios were comparable in our 'per protocol' analysis. Similarly, there was only weak evidence for an effect of phase on incidence of outbreaks in both intention-to-treat and 'per protocol' analyses, with wide confidence intervals inclusive of no difference in incidence (*Table 3*).

We compared HOCI-level impacts of the sequence report between phases. Nosocomial linkage to other individual cases, where initial IPC investigation had not correctly identified any such linkage, was identified in 6.7 and 6.8% of all HOCI cases in the rapid and longer-turnaround phases, respectively (OR for 'rapid vs. longer-turnaround' 0.98, 95% CI 0.46–2.08; p=0.95) (*Table 2*) and in 11.4 and 12.6% respectively of cases where the report was returned. For 25 cases in the rapid and 5 cases in the longer-turnaround phase, phylogenetic trees were used for sequences with <90% genome coverage, with 3 from the rapid phase showing previously unidentified linkage.

IPC practices were changed in 7.8 and 7.4% of all HOCI cases in the rapid and longer-turnaround phases, respectively (OR for 'rapid vs. longer-turnaround' 1.07, 0.34–3.38; p=0.90), and 17.2 and 11.6%, respectively, of cases where the report was returned. No one specific change to IPC action dominated those recorded among the options included within study reporting forms (*Appendix 1— table 2*, *Appendix 1—table 3*). When restricted to higher performing sites (i.e. 'per protocol'), IPC practice was changed in a greater proportion of all HOCI cases in the rapid (9.9%) in comparison to the longer-turnaround (0.7%) sequencing phase (OR for 'rapid vs. longer-turnaround' 15.55, 1.30– 1.85; p=0.01) and 16.7 and 1.1%, respectively, of cases where SRT reports were returned. The impact of phase on detecting nosocomial linkage was similar.

IPC teams more commonly reported finding the sequence reports useful in the rapid sequencing, 303/428 (70.8%) compared to the longer-turnaround phase, 107/215 (49.8%) (although this association was reversed on analysis within the multi-level mode specified, OR 0.82 rapid vs. longer-turnaround, 0.12–5.46; p=0.82), and the difference was more pronounced in the 'per protocol' analysis (79.0 vs. 27.2%, respectively; OR 3.44, 0.28–42.61; p=0.41). We explored this association further using the actual time to return of the reports, going beyond the analyses pre-specified in the SAP (*Figure 2*). In the 'per protocol' analysis, an impact on IPC actions was observed in 20.7% (45/217) of HOCI cases in which the SRT report was returned within 5 days, but in very few cases beyond this, with this trend less

**Table 2.** Per hospital-onset COVID-19 infection (HOCI) implementation and outcome summary by study intervention phase, overall and within the 7/14 sites included in the 'per protocol' sensitivity analysis.

| | All study sites | | | Sensitivity analysis | |
|---|---|---|---|---|---|
| | Study phase | | | Study phase | |
| | Longer-turnaround | Rapid | Total | Longer-turnaround | Rapid |
| N HOCI cases | 373 | 947 | 1320 | 143 | 533 |
| *Implementation* | | | | | |
| Sequence returned within expected timeline, n (%)* | 229 (61.4) | 377 (39.8) | 606 (45.9) | 81 (56.6) | 204 (38.3) |
| Sequence returned within study period, n (%)* | 277 (74.3) | 596 (62.9) | 873 (66.1) | 98 (68.5) | 347 (65.1) |
| SRT report returned within target timeline (10 days for longer-turnaround, 2 days for rapid), n (%) | 79 (21.2) | 44 (4.6) | 123 (9.3) | 35 (24.5) | 44 (8.3) |
| SRT report returned within study period, n (%) | 215 (57.6) | 435 (45.9) | 650 (49.2) | 92 (64.3) | 317 (59.5) |
| Time from sample to report return (days), median (IQR, range) [n] | 13 (9–15, 0–36) [215] | 5 (3-11, 2-84) [430] | 9 (4-14, 0-84) [645] | 13 (9–17, 6–29) [92] | 4 (3-6, 2-64) [312] |
| *Sequencing results* | | | | | |
| SRT-suggestive patient acquired infection post-admission, n/N (%) | 196/212 (92.5) | 384/423 (90.8) | 580/635 (91.3) | 85/92 (92.4) | 287/311 (92.3) |
| SRT-suggestive patient is part of ward outbreak, n/N (%) | 151/212 (71.2) | 260/423 (61.5) | 411/635 (64.7) | 65/92 (70.7) | 202/311 (65.0) |
| *Linkage identified not suspected at initial IPC investigation:* | | | | | |
| All HOCIs in phase n/N (%[†], 95% CI) | 24/348 (6.8, 1.7–11.8) | 46/915 (6.7, 2.0–11.3) | 70/1263 (5.5) | 11/139 (7.9, 3.4–12.4) | 39/512 (7.6, 5.3–9.9) |
| When SRT returned n/N (%) | 24/190 (12.6) | 46/403 (11.4) | 70/593 (11.8) | 11/88 (12.5) | 39/296 (13.2) |
| SRT excluded IPC-identified hospital outbreak, n/N (%) | 14/213 (6.6) | 27/428 (6.3) | 41/641 (6.4) | 9/92 (9.8) | 25/310 (8.1) |
| *Impact on IPC* | | | | | |
| *SRT changed IPC practice:* | | | | | |
| All HOCIs in phase n/N (%[†], 95% CI) | 25/373 (7.4, 1.1–13.6) | 74/941 (7.8, 2.4–13.2) | 99/1314 (7.5) | 1/143 (0.7, 0.0–2.1) | 52/527 (9.9, 7.3–12.4) |
| When SRT returned n/N (%) | 25/215 (11.6) | 74/429 (17.2) | 99/644 (15.4) | 1/92 (1.1) | 52/311 (16.7) |
| SRT changed IPC practice for ward, n/N (%) | 13/215 (6.0) | 31/429 (7.2) | 44/644 (6.8) | 0/92 (0.0) | 28/311 (9.0) |
| SRT used in IPC decisions beyond ward, n/N (%) | 12/215 (5.6) | 45/428 (10.5) | 57/643 (8.9) | 1/92 (1.1) | 27/310 (8.7) |
| *IPC team reported SRT to be useful, n/N (%)* | | | | | |
| Yes | 107/215 (49.8) | 303/428 (70.8) | 410/643 (63.8) | 25/92 (27.2) | 245/310 (79.0) |
| No | 67/215 (31.2) | 71/428 (16.6) | 138/643 (21.5) | 50/92 (54.3) | 57/310 (18.4) |
| Unsure | 41/215 (19.1) | 54/428 (12.6) | 95/643 (14.8) | 17/92 (18.5) | 8/310 (2.6) |

*Table 2 continued on next page*

Table 2 continued

| | All study sites | | | Sensitivity analysis | |
| | Study phase | | | Study phase | |
| | Longer-turnaround | Rapid | Total | Longer-turnaround | Rapid |
|---|---|---|---|---|---|
| **HCW absence on ward** | | | | | |
| Proportion of HCWs on sick leave due to COVID-19, median (IQR, range) [n] | 0.09 (0.00–0.15, 0.00–0.30) [49] | 0.13 (0.07–0.29, 0.00–1.00) [162] | 0.13 (0.04–0.27, 0.00–1.00) [321] ‡ | 0.09 (0.00–0.15, 0.00–0.30) [49] | 0.13 (0.08–0.29, 0.00–1.00) [143] |

HCW, healthcare worker; HOCI, hospital-onset COVID-19 infection; IPC, infection prevention and control; IQR, interquartile range; SRT, sequence reporting tool.

*As recorded by site, not based on recorded date or availability on central CLIMB server.

†Estimated marginal value from mixed effects model, not raw %, evaluated on intention-to-treat basis with lack of SRT report classified as 'no'.

‡Includes data for baseline phase: 0.13 (0.00–0.30, 0.00–0.88) [110].

apparent when data from all sites were considered. *Figure 2* also displays a strong decline in reported usefulness of the SRT with increasing turnaround time, both across all sites and in the 'per protocol' analysis. Sequence reports were considered useful in 79.1% (182/230) of cases if returned within 5 days for all sites (169/216, 78%, in 'per protocol' analysis). However, we note that many of the HOCI cases with SRT returned within 5 days were from a single study site, and some sites did not seem to

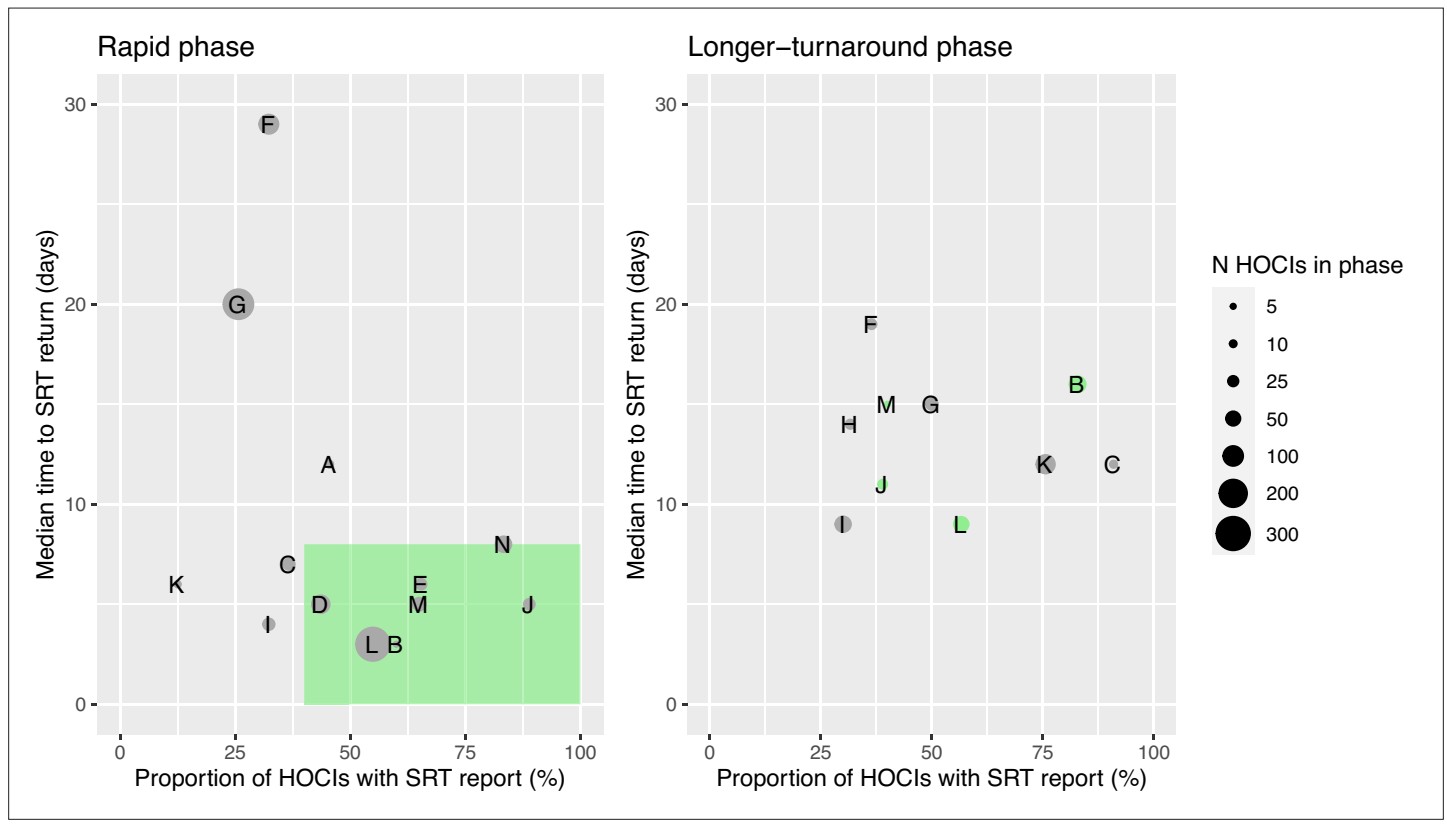

**Figure 1.** Plots of the median turnaround time against the percentage of hospital-onset COVID-19 infection (HOCI) cases with sequence reporting tool (SRT) reports returned for the rapid (left panel) and longer-turnaround (right panel) sequencing phases across the 14 study sites. The size of each circle plotted indicates the number of HOCI cases observed within each phase for each site, with letter labels corresponding to study site. The criteria for inclusion in our sensitivity analysis are displayed as the green rectangle in the rapid phase plot, and sites on the longer-turnaround phase plot are colour-coded by their inclusion. In the rapid phase, SRT reports were returned for 0/4 HOCI cases recorded for site H. Site N did not have a longer-turnaround phase, Site A observed 0 HOCI cases, and sites D and E returned SRT reports for 0/1 and 0/2 HOCI cases, respectively, in this phase.

**Table 3.** Incidence outcomes by study intervention phase, overall and within the 7/14 sites included in the 'per protocol' sensitivity analysis.

| | Study phase | | | IRR† (95% CI, p-value) | |
| --- | --- | --- | --- | --- | --- |
| | Baseline | Longer-turnaround | Rapid | Longer-turnaround vs. baseline | Rapid vs. baseline |
| *All sites* | | | | | |
| n HOCI cases | 850 | 373 | 947 | – | – |
| n IPC-defined HAIs | 488 | 207 | 576 | – | – |
| Weekly incidence of IPC-defined HAIs per 100 inpatients, mean (median, IQR, range)* [primary outcome] | 1.0 (0.5, 0.0–1.4, 0.0–5.6) | 0.7 (0.3, 0.0–0.7, 0.0–7.6) ‡ | 0.6 (0.3, 0.0–0.8, 0.0–5.3) ‡ | 1.60 (0.85–3.01; 0.14) | 0.85 (0.48–1.50; 0.54) |
| n IPC-defined outbreak events | 129 | 33 | 114 | – | – |
| Weekly incidence of IPC-defined outbreak events per 1000 inpatients, mean (median, IQR, range)* | 2.7 (1.1, 0.0–4.1, 0.0–23.0) | 0.8 (0.0, 0.0–1.0, 0.0–8.9) ‡ | 0.7 (0.0, 0.0–0.0, 0.0–8.9) ‡ | 1.09 (0.38–3.16; 0.86) | 0.58 (0.24–1.39; 0.20) |
| n IPC + sequencing-defined outbreak events | – | 40 | 133 | – | – |
| Weekly incidence of IPC + sequencing-defined outbreak events per 1000 inpatients, mean (median, IQR, range)* | – | 1.1 (0.0, 0.0–1.5, 0.0–13.4) ‡ | 0.9 (0.0, 0.0–1.4, 0.0–7.6) ‡ | – | – |
| *Sensitivity analysis* | | | | | |
| n HOCI cases | 290 | 143 | 533 | – | – |
| n IPC-defined HAIs | 179 | 91 | 337 | – | – |
| Weekly incidence of IPC-defined HAIs per 100 inpatients, mean (median, IQR, range)* [primary outcome] | 0.3 (0.0, 0.0–0.3, 0.0–3.0) | 0.3 (0.0, 0.0–0.0, 0.0–3.4) ‡ | 0.4 (0.0, 0.0–0.3, 0.0–5.3) ‡ | 2.21 (0.82–5.92; 0.10) | 1.75 (0.75–4.08; 0.16) |
| n IPC-defined outbreak events | 58 | 14 | 55 | – | – |
| Weekly incidence of IPC-defined outbreak events per 1000 inpatients, mean (median, IQR, range)* | 1.1 (0.0, 0.0–1.3, 0.0–12.9) | 0.3 (0.0, 0.0–0.0, 0.0–5.7) ‡ | 0.4 (0.0, 0.0–0.0, 0.0–8.9) ‡ | 0.83 (0.14–4.93; 0.80) | 0.46 (0.11–1.86; 0.21) |
| n IPC + sequencing-defined outbreak events | – | 14 | 67 | – | – |
| Weekly incidence of IPC + sequencing-defined outbreak events per 1000 inpatients, mean (median, IQR, range)* | – | 0.3 (0.0, 0.0–0.0, 0.0–5.7) ‡ | 0.5 (0.0, 0.0–0.0, 0.0–7.6) ‡ | – | – |

IPC-defined HAIs are considered to be 'probable' or 'definite' HAIs.

HAI, hospital-acquired infection; HOCI, hospital-onset COVID-19 infection; IPC, infection prevention and control; IQR, interquartile range; IRR, incidence rate ratio.

*Descriptive data over all week-long periods at all study sites.

†Adjusted for proportion of current inpatients at site that are COVID-19 cases, community incidence rate, and calendar time (as displayed in *Appendix 1—figure 5* and *Appendix 1—figure 6* for all sites).

‡Not including data from the first week of each intervention period or in the week following any break in the intervention period.

have clearly differentiated 'useful' SRT reports when completing data collection (*Appendix 1—figure 2* and *Appendix 1—figure 3*).

SRT reports suggested that 91.3% of HOCI patients had acquired their infection post-admission (580/635, *Table 2*). In 91.9%, (589/641, *Appendix 1—table 2*) of cases, the reports were interpreted as supportive of IPC actions already taken. SRT reports also suggested post-admission infection in the majority of indeterminate HAIs (diagnosed 3–7 days from admission) (176/223, 78.9%).

Our analysis models reveal important findings beyond the effect of the intervention. The analysis model for the incidence of HAIs identified independent positive associations with the proportion of current SARS-CoV-2-positive inpatients, the local community incidence of new SARS-CoV-2 cases (which peaked from December 2020 to January 2021, *Appendix 1—figure 4* and *Appendix 1—figure 5*), and calendar time (modelled as 'study week'). Adding the proportion of local community cases that were Alpha (lineage B.1.1.7) variant did not lead to a statistically significant improvement in

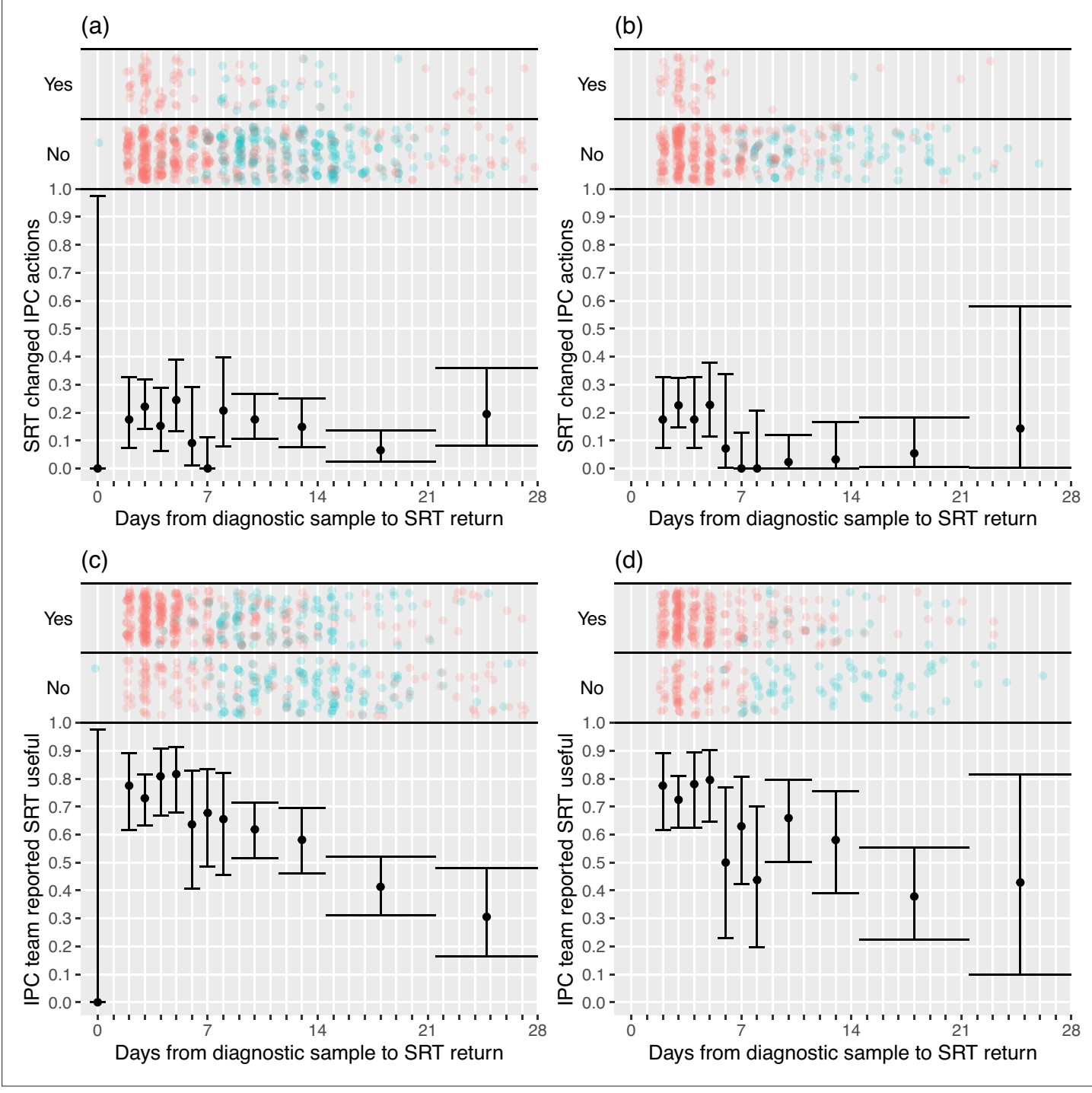

**Figure 2.** Plots of the proportion of returned sequence reporting tool (SRT) reports that had an impact on infection prevention and control (IPC) actions (**a**, **b**) and that were reported to be useful by IPC teams (**c**, **d**). Data are shown for all sites in (**a**) and (**c**), and for the seven sites included in the 'per protocol' sensitivity analysis in (**b**) and (**d**). Results are only shown up to turnaround times of ≤28 days, and grouped proportions are shown for ≥9 days because of data sparsity at higher turnaround times. Error bars show binomial 95% CIs. 'Yes' and 'No' outcomes for individual hospital-onset COVID-19 infection (HOCI) cases are displayed, colour-coded by rapid (red) and longer-turnaround (blue) intervention phases and with random jitter to avoid overplotting. 'Unsure' responses were coded as 'No' for (**c**) and (**d**).

model fit (p=0.78). The observed weekly HOCI incidence rates varied substantially from 0 to 7.6 per 100 SARS-CoV-2 negative inpatients, with peaks aligning with those for local community incidence (*Appendix 1—figure 4*).

From modelling outbreaks, positive associations were similarly found for both hospital prevalence and community incidence of SARS-CoV-2 (*Appendix 1—figure 6*). The median number of HOCIs per IPC-defined outbreak event was four, with the largest observed outbreak including 43 HOCIs (*Appendix 1—table 1*).

Extensive qualitative analyses (Mapp et al., under review; *Flowers et al., 2022*) found high levels of acceptability for the SRT sequencing reports, which supported decision-making about IPC activity (e.g. stand down some IPC actions or continue as planned). In several sites, the major barriers to embedding and normalising the SRT within existing systems and processes were overcome. The SRT did provide new and valued insights into transmission events, outbreaks, and wider hospital functioning but mainly acted to offer confirmation and reassurance to IPC teams. Critically, given the context of the study within the pandemic timeline, the capacity to generate and respond to these insights effectively on a case-by-case basis was breached in most sites by the volume of HOCIs, and the limits of finite human and physical resource (e.g. hospital layout).

## Cost of SARS-CoV-2 genome sequencing

Analysis of SARS-CoV-2 genome sequencing in the 10 laboratories which performed the tests for the sites included in the study showed that mean per-sample costs were on average higher for rapid (£78.11) vs. longer-turnaround (£66.94) sequencing. (*Appendix 1—table 4*). Consumables were the highest cost driver of the sequencing process accounting for 66% in rapid and 67% in longer-turnaround sequencing.

Several factors affected the costs of genome sequencing. There was a general tendency of increasing returns to scale, with average per-sample costs of genome sequencing tending to decrease as the batch size increases; cost per sample in reagents also depends highly on how many samples are processed per batch. Another factor was the sequencing platform and protocols used: some processes had been automated which reduced the hands-on input.

## Discussion

This study constitutes the largest prospective multicentre evaluation study of viral whole-genome sequencing (WGS) for acute IPC investigation of nosocomial transmission conducted to date. The study was run as part of routine practice within the NHS, and the challenges faced in implementing the intervention reflected the context and barriers in winter 2020–2021 in the UK. We did not demonstrate a direct impact of sequencing on the primary outcome of the incidence of HAIs, either on full analysis or when restricted to the higher performing sites, and the overall proportion of cases with nosocomial transmission linkage identified using sequencing that had been missed by IPC investigation was <10% in the intervention phases. However, on post hoc exploratory investigation among those sites with the most effective implementation of the sequencing intervention we showed that feedback within 5 days of diagnosis allowed for maximal impact on IPC actions. IPC teams, particularly in the 'per protocol' analysis, were almost all positive in their perception of the utility of viral sequencing for outbreak investigation.

The study was undertaken during a period of extreme strain on the NHS, with hospitals described as being 'in the eye of a COVID-19 storm' (*Oliver, 2021a*). Sites reported that they lacked the additional resources, in terms of staff and bed space, needed to respond effectively to insights generated by sequencing. Furthermore, if the study were repeated now then IPC teams would have more evidence-backed tools at their disposal, such as increasing respirator usage. As such, we do not believe that the null result for the impact on incidence of nosocomial transmission should be taken as strong evidence for a general lack of effectiveness of viral WGS for IPC.

Outbreak investigations are inherently complex and must take account of uncertainty regarding transmission links, even in the presence of high-quality genomic data (*Lindsey et al., 2022*). Interventions centred on IPC practices often need to be evaluated at the hospital level in order to allow for impacts on transmission across an institution as a whole (*O'Hara et al., 2019*), meaning that large multicentre studies are required to generate high-quality evidence. Standardisation of data collection

with complex structures across multiple hospital sites is a considerable challenge. A review of IPC practice guidelines conducted prior to the SARS-CoV-2 pandemic found that most recommendations were based on evidence from descriptive studies, expert opinion, and other low-quality evidence (*Mitchell et al., 2020*).

The use of viral WGS for public health surveillance has become firmly established in the UK for SARS-CoV-2 (*The COVID-19 Genomics UK consortium, 2020*). This enabled early detection of the increased transmissibility and health impact of the Alpha variant (*Volz et al., 2021*) and subsequent monitoring of the Delta (*Mishra et al., 2021*) and Omicron variants (*Andrews et al., 2022*). However, whilst viral WGS for acute outbreak investigation has been shown for both SARS-CoV-2 (*Meredith et al., 2020*; *Snell et al., 2022b*; *Lumley et al., 2021*; *Li et al., 2021*) and other viruses (*Brown et al., 2019*; *Houldcroft et al., 2018*; *Roy et al., 2019*) to better identify sources of hospital-acquired infections and transmission chains, its impact on the management and outcome of nosocomial infection has not previously been quantified. Our study provides a substantial body of evidence regarding the introduction of viral WGS into routine IPC practice, its potential usage for outbreak management, and the challenges that need to be overcome to achieve implementation across the UK.

There are several limitations that may have impacted the results of this study. The study was conducted between October 2020 and April 2021. In this period, the local community incidence for the study sites ranged from <50 to >1200 weekly cases per 100,000 people. There were corresponding large variations in the healthcare burden of COVID-19, with several sites recording weeks when more than half of all inpatients were SARS-CoV-2 positive. High community infection rates and associated increases in the incidence of HOCI cases contributed to difficulties for site research teams in generating good quality viral sequences and reports for all HOCI cases within target time frames.

Our qualitative analyses also found that the capacity of sites to react to information generated by the sequencing intervention was breached by the volume of HOCI and admitted COVID-19 cases ("we've been basically deluged", IPC staff) in combination with the finite personnel resources and limited physical space for isolation that was available ("The trouble is when you have so many wards going down and such a high prevalence of COVID, your actions are kind of the same regardless", IPC staff). It may therefore be more achievable to develop effective systems for rapid viral WGS and feedback for endemic respiratory viruses at lower and more consistent levels, and more timely reporting of results might be associated with greater impact on IPC actions. As well as acute changes to IPC actions, there is the potential for routine pathogen sequencing to allow prospective IPC practice and policies to be refined. This could enable a longer-term reduction in the incidence of nosocomial infection at any given site, and such effects would be less dependent on turnaround time of sequencing in any given case. However, the capacity of sites to make such informed adjustments to IPC practice was limited during peaks in incidence of SARS-CoV-2 over the time scale of this study.

Planning this study and developing the data collection forms during the early stages of a novel viral pandemic was challenging, as in the summer of 2020 there were still ongoing debates around the primary mode of viral transmission and optimal IPC practice, and global supply chains for personal protective equipment were strained. In the planning of an equivalent study now, there would be a greater focus on adjustments to ventilation (*Allen and Ibrahim, 2021*), air filtration (*Conway Morris et al., 2022*), and respirator (*Ferris et al., 2021*) usage. It would also be possible to be more prescriptive and standardised regarding the recommended changes to IPC practice in response to sequencing findings, with the potential that our improved knowledge and available tools might facilitate a measurable impact on the incidence of nosocomial transmission.

The peak in SARS-CoV-2 levels from December 2020 to January 2021 corresponded to the rise of the highly transmissible Alpha variant in the UK (*Volz et al., 2021*). We did not find that the local prevalence of the Alpha variant was associated with the incidence rate of HAIs, beyond any effect mediated by higher community incidence. This matches the conclusions of a previously reported sub-study analysis using data from our sites (*Boshier et al., 2021*).

The study intervention made use of a bespoke SRT (*Stirrup et al., 2021*). The SRT combined both patient metadata and sequencing data, providing a single-page, easily interpretable report for IPC teams. It also facilitated standardisation of data collection across sites. Interestingly, while

HOCIs diagnosed 3–7 days after admission are generally excluded from assessments of noso-comial SARS-CoV-2 infections (*Oliver, 2021b*), because of difficulty in distinguishing them from community-acquired infections, the SRT reported the majority (78.9%) of these indeterminate HAIs as being hospital-acquired. This confirms findings from a retrospective study using genomic linkage (*Lumley et al., 2021*), and may reflect a shorter incubation time for the Alpha variant compared to earlier variants (*Snell et al., 2022a*; although this remains uncertain; *Blanquart et al., 2022*), indicating that definitions used for monitoring and reporting may need to be kept under active review. Variants with shorter incubation times would lead to a greater importance for the rapidity of feedback in informing adjustments to IPC actions.

A number of limitations of the SRT were recognised, and work is ongoing to rectify these for future studies. The SRT's probability calculations did not include patient and HCW movements. The SRT gave feedback on cases that could plausibly form part of the same outbreak but did not identify direct transmission pairs or networks, as has been done in other studies (*Lindsey et al., 2022*; *Illingworth et al., 2022*). HCW sequencing data could not be incorporated at all sites due to logistical and data management and access constraints. Implementation of an improved tool with these features might help to better identify routes of transmission within a hospital that could be interrupted, for example, through changes to the management of ward transfers for patients, isolation policies or identification of areas within the hospital linked to high risk of transmission. Finally, samples with less than 90% genome coverage were not included within the reporting system, despite the fact that they may still be useful for phylogenetic analyses.

The study sites varied in their ability to process sequence and metadata and generate and distribute reports in a timely manner (*Figure 1*), and the targeted turnaround times for reporting were not achieved at any of the sites for the majority of HOCIs in either the 'rapid' or 'longer turnaround' phases. Sites that had established teams with existing genomics expertise and on-site sequencing facilities were generally more successful at implementing the SRT into clinical practice (Mapp et al., under review). There is a need to focus on how sequencing and reporting processes can be integrated within local infrastructure and tailoring of local processes to ensure clear chains of communication from diagnostic labs through to the IPC team. Precisely understanding the barriers to achieving rapid turnaround times is key to future IPC use of viral WGS and is currently being analysed in a follow-up secondary analysis. Standardising and automating more of the SRT production pipeline will also help reduce the implementation burden at sites.

The study covered a period in which a national vaccination programme was initiated for HCWs and the elderly population in the UK, commencing with those ≥80 years from 8 December 2020. We had planned to include data on the proportion of HCWs who had received at least one vaccine dose as a variable in the analysis of incidence outcomes. This was subsequently not included because data was only available from 10 sites, for which rollout of HCW vaccination was broadly consistent. As such, any effect of HCW or patient vaccination on the incidence outcomes would form part of the estimated association with calendar time.

With the sequencing technology now available and high levels of interest in viral genomics for public health, there is the potential to incorporate viral WGS into routine IPC practice. Many publications have already highlighted the utility of viral sequence data for changing IPC policy and auditing the management of outbreaks (*Meredith et al., 2020*; *Snell et al., 2022b*; *Lumley et al., 2021*; *Li et al., 2021*; *Brown et al., 2019*; *Houldcroft et al., 2018*; *Roy et al., 2019*). We did not demonstrate an effect of our sequencing intervention on our primary outcome of the incidence of HAIS, and there were challenges in the implementation of the intervention. However, our study provides the first prospective evidence that with faster turnaround times, viral sequences can inform ongoing IPC actions in managing nosocomial infections; on post hoc exploratory analysis results returned within ≤5 days from sampling to result changed the actions of IPC teams in around 20% of cases. The SRT, by rapidly combining sequence and patient metadata, was also better able than standard IPC definitions alone to distinguish hospital and community-acquired infections within a clinically relevant time scale. The difference in the cost of rapid compared with longer-turnaround hospital sample sequencing is low relative to the overall cost level at present (*Appendix 1—table 4*). Assuming SARS-CoV-2 sequencing for public health purposes continues, the added cost of rapid sequencing for IPC purposes could potentially be offset by the benefits accrued.

While we did not show an impact of sequencing on the numbers of HAIs or outbreaks, the evidence that these correlated with the high community SARS-CoV-2 rates suggests that factors beyond the control of IPC were influential. Our study nonetheless provides valuable evidence regarding the implementation and utility of this technology for IPC, and potentially it will have a greater positive impact on IPC practice outside of the burdens and resource constraints imposed by a pandemic. Importantly for future research, we provide a wealth of data on why the study worked better at some sites than others, and the challenges that would need to be overcome to make full use of viral genome sequencing for IPC practice more widely. It remains to be demonstrated that viral sequencing can have a direct impact on clinical outcomes such as the incidence of HAIs, and further prospective studies with refined implementation of similar interventions are required to address this.

## Acknowledgements

COG-UK is supported by funding from the Medical Research Council (MRC) part of UK Research & Innovation (UKRI), the National Institute of Health Research (NIHR) (grant code: MC_PC_19027), and Genome Research Limited, operating as the Wellcome Sanger Institute. UCLH APDU is funded by the UCLH NHS Foundation Trust and the UCLH NIHR BRC. MRC-University of Glasgow Centre for Virus Research has received funding from the Medical Research Council. SR is part-funded by the Research England's Expanding Excellence in England (E3) Fund, and additional funding for sequencing of SARS-CoV-2 at University of Portsmouth came from the Wessex Academic Health Sciences Centre (AHSC). FC was funded by a Wellcome Trust Sir Henry Postdoctoral Fellowship. LS has received support for research from NIHR Biomedical Research Centre at Guy's and St Thomas NHS Foundation Trust, Guy's & St Thomas Charity and King's Health Partners. We also acknowledge the support of the independent members of the Joint Trial Steering Committee and Data Monitoring Committee (TSC-DMC): Prof Marion Koopmans (Erasmus MC), Prof Walter Zingg (University of Geneva), Prof Colm Bergin (Trinity College Dublin), Prof Karla Hemming (University of Birmingham), and Prof Katherine Fielding (LSHTM). Also, TSC-DMC non-independent members: Prof Nick Lemoine (NIHR CRN) and Prof Sharon Peacock (COG-UK). We also thank the members of COG-UK who have directly supported the study: Dr Ewan Harrison (Cambridge University), Dr Katerina Galai (PHE), Dr Francesc Coll (LSHTM), Dr Michael Chapman (HDR-UK), Prof Thomas Connor and team (Cardiff University), and Prof Nick Loman and team (University of Birmingham). We also thank the COG-UK Consortium and the UK National Institute for Health Research Clinical Research Network (NIHR CRN).

## Additional information

### Competing interests

Gee Yen Shin: has an unpaid role as Deputy Chair, British Medical Association London Regional Council. The author has no other competing interests to declare. Maria-Teresa Cutino-Moguel: received payment for anonymous interview conducted by Adkins Research Group. The author has no other competing interests to declare. Eleni Nastouli: holds grants by NIHR, EPSRC, MRC-UKRI, H2020, ViiV Healthcare, Pfizer and Amfar, and has received grants to attend meetings from H2020 and ViiV Healthcare. Darren Smith: holds the following grants that are not specifically for the present work: COG-UK, PHE test and trace funded the sequencing aspect. HOCI funded a technician to support sequencing during study period. The author has no other competing interests to declare. Francesc Coll: received consulting fees from Next Gen Diagnostics LLC (during 2018/2019), received payment or honoria for lectures from University of Cambridge and Wellcome Genome Campus Advanced Courses, and received support for attending meeting and/or travel to meetings from European Congress of Clinical Microbiology & Infectious Diseases (ECCMID), The American Society for Microbiology (ASM), Microbiology Society, European Congress of Clinical Microbiology & Infectious Diseases (ECCMID), and the British Infection Association (BIA). The author has no other competing interests to declare. Sharon J Peacock: received consultancy fees from Pfizer (Coronavirus External Advisory Board) and Melinta Therapeutics, received payment from SVB Leerink for a round table meeting and for Mary Strauss Distinguished Public Lecture from the Fralin Biomedical Research Institute, US, and

support for attending ICPIC conference, Geneva and World Health Summit, Berlin in 2021, and hold stocks or stock options in Specific Technologies (European Union Scientific Advisory Board) and Next Gen Diagnostics (Scientific Advisory Board). SP also serves as Chair, Medical Advisory Committee, Sir Jules Thorn Charitable Trust, Board member of the Wellcome SEDRIC (Surveillance and Epidemiology of Drug Resistant Consortium), and Non-Executive Director of Cambridge University Hospitals NHS Foundation Trust. The author has no other competing interests to declare. COG-UK HOCI Investigators: The COVID-19 Genomics UK (COG-UK) consortium: Judith Breuer: is a member of the SAGE hospital onset covid working group 2020-2022. The author has no other competing interests to declare. The other authors declare that no competing interests exist.

## Funding

| Funder | Grant reference number | Author |
|---|---|---|
| Medical Research Council | | Judith Breuer |
| National Institute for Health and Care Research | MC_PC_19027 | Judith Breuer |

The funders had no role in study design, data collection and interpretation, or the decision to submit the work for publication.

## Author contributions

Oliver Stirrup, Conceptualization, Data curation, Software, Formal analysis, Validation, Investigation, Visualization, Methodology, Writing - original draft, Writing – review and editing; James Blackstone, Conceptualization, Data curation, Funding acquisition, Investigation, Methodology, Project administration, Writing – review and editing; Fiona Mapp, Conceptualization, Methodology, Writing – review and editing; Alyson MacNeil, Data curation, Project administration, Writing – review and editing; Monica Panca, Formal analysis, Investigation, Methodology, Writing – review and editing; Alison Holmes, Paul Flowers, Andrew Copas, Conceptualization, Supervision, Methodology, Writing – review and editing; Nicholas Machin, Gee Yen Shin, Tabitha Mahungu, Kordo Saeed, Tranprit Saluja, Yusri Taha, Nikunj Mahida, Cassie Pope, Anu Chawla, Maria-Teresa Cutino-Moguel, David L Robertson, Eleni Nastouli, Supervision, Investigation, Writing – review and editing; Asif Tamuri, Software, Methodology, Writing – review and editing; Rachel Williams, Flavia Flaviani, Samuel Robson, Darren Smith, Matthew Loose, Kenneth Laing, Irene Monahan, Beatrix Kele, Sam Haldenby, Ryan George, Matthew Bashton, Adam A Witney, Matthew Byott, Gaia Nebbia, Luke B Snell, Data curation, Investigation, Writing – review and editing; Alistair Darby, Data curation, Supervision, Investigation, Writing – review and editing; Francesc Coll, Michael Chapman, Conceptualization, Writing – review and editing; Sharon J Peacock, Conceptualization, Funding acquisition, Writing – review and editing; COG-UK HOCI Investigators, The COVID-19 Genomics UK (COG-UK) consortium, Investigation; Joseph Hughes, Matthew Parker, Sunando Roy, Data curation, Investigation, Methodology, Writing – review and editing; David G Partridge, Christine Peters, Thushan I de Silva, Emma Thomson, Conceptualization, Supervision, Investigation, Writing – review and editing; James Richard Price, Conceptualization, Investigation, Writing – review and editing; Judith Breuer, Conceptualization, Supervision, Funding acquisition, Investigation, Methodology, Project administration, Writing – review and editing

## Author ORCIDs

Oliver Stirrup ⓘ http://orcid.org/0000-0002-8705-3281
James Blackstone ⓘ http://orcid.org/0000-0003-4335-5269
Alyson MacNeil ⓘ http://orcid.org/0000-0001-8409-2755
David L Robertson ⓘ http://orcid.org/0000-0001-6338-0221
Flavia Flaviani ⓘ http://orcid.org/0000-0002-4210-0451
Sharon J Peacock ⓘ http://orcid.org/0000-0002-1718-2782
David G Partridge ⓘ http://orcid.org/0000-0002-0417-2016
Judith Breuer ⓘ http://orcid.org/0000-0001-8246-0534

## Ethics

Clinical trial registration ClinicalTrials.gov Identifier: NCT04405934.
Human subjects: Ethical approval for the study was granted by NHS HRA (REC 20/EE/0118). The need for consent from individual participants was waived because the study involved a hospital-level

intervention that did not directly affect the clinical management of individual participants once diagnosed with a SARS-COV-2 infection.

## Decision letter and Author response
Decision letter https://doi.org/10.7554/eLife.78427.sa1
Author response https://doi.org/10.7554/eLife.78427.sa2

## Additional files

### Supplementary files
• Supplementary file 1. List of COG-UK HOCI investigators.
• Supplementary file 2. Member list for the COVID-19 Genomics UK (COG-UK) consortium.
• MDAR checklist

### Data availability
A fully anonymised version of the dataset generated and analysed for this study is available on the UCL Research Data Repository (https://doi.org/10.5522/04/20769637.v1).

The following dataset was generated:

| Author(s) | Year | Dataset title | Dataset URL | Database and Identifier |
|---|---|---|---|---|
| Oliver S, James B, Andrew C, Judith B | 2022 | COG-UK hospital-onset COVID-19 infection study dataset | https://doi.org/10.5522/04/20769637.v1 | Dyrad Digital Repository, 10.5522/04/20769637.v1 |

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

# Appendix 1

Appendix 1 contains further details of statistical analysis methods, and supplementary figures and tables describing additional study data.

## Methods

### Sample size estimation

There was uncertainty in the number of HOCIs that would be identified at each site during each of the intervention periods, with the rapid sequencing phase being 8 weeks' duration. We assumed that there may be an average of 10 HOCIs/week per site during this intervention period, a total of 80 per site. Within a typical site this would allow us to estimate the proportion of HOCIs with genotypic linkage to another case(s) not detected by IPC processes with minimum precision of ±9.4%. Similarly, we would be able to estimate the proportion of HOCIs where an action is taken that would not have occurred without sequencing within ±9.4%, with a pooled estimate of key proportions across the 14 sites implementing rapid sequencing within ±6.5% assuming an intra-cluster correlation coefficient of 0.05.

Comparing the proportion of HOCIs with genotypic linkage to another case(s) not detected by IPC processes between rapid testing and delayed testing phases across all sites, the study aimed for at least 80% power to detect a percentage point difference of 11% (two-sided test with alpha = 0.05, considering proportions of 55.5% vs. 44.5% which would be associated with minimum power for a difference of this magnitude).

For the outcome of weekly incidence of IPC-defined HOCIs, using an approximate normal distribution for weekly counts there was 86.7% power to demonstrate a reduction from 12 IPC-defined HOCIs per week in the baseline phase to 10 per week during the rapid testing phase across all sites, under 5% significance level two-tailed testing. However, these calculations correspond to a variance of 12 for weekly counts based on the Poisson distribution, but the presence of over-dispersion of weekly counts would lead to a lower power to detect a difference. Using an overdispersion parameter of 0.82 based on retrospective analysis of data from Sheffield and Glasgow (dataset as described by *Stirrup et al., 2021*) resulted in 81% power to detect a reduction in mean weekly incidence from 12.5 to 10.

## Planned secondary outcomes dropped from formal analysis

We did not carry out formal statistical analysis for the following planned secondary outcomes:

- Weekly incidence of IPC + sequencing-defined SARS-CoV-2 hospital outbreaks, measured as incidence rate per week per 100 non-COVID-19 inpatients, during each phase of the study based on case report forms.
- The number of HCW periods of sickness/self-isolation as assessed as a proportion of the number of staff usually on those wards impacted by HOCI cases, for all phases of the study.

The first of these was dropped prior to analysis because of incomplete sequencing coverage of HOCI cases in the intervention phases – it was not felt that this would add useful information given the level of sequencing achieved and the null results for other incidence outcomes. The second was dropped (again prior to any statistical analysis) because of low levels of data completion at most of the study sites. Data collection on HCW absence was discontinued whilst the study was ongoing in order to reduce the administrative burden for sites and due to difficulties in accessing this data for research staff (with staff data being recorded on separate systems to patient data).

We also omitted specific reporting of the secondary outcome of 'Ideal changes to IPC actions following receipt of sequencing report'. This was because no recommended changes to IPC actions were recorded that were also recorded as 'not implemented'. As such, this outcome was identical to 'changes to IPC actions following receipt of sequencing report'.

## Coding of primary and secondary outcomes

### Primary outcome 1

#### Incidence of IPC-defined SARS-CoV-2 HAIs

In order to standardise this measure across sites, 'IPC-defined SARS-CoV-2 HAIs' were considered to be all HOCIs with an interval of ≥8 days from admission to symptom onset (if known) or sample date (i.e. those meeting the PHE definition of a probable or definite HAI; *Public Health England, 2020*).

Incidence was expressed 'per 100 non-COVID-19 inpatients per site per week' and was evaluated for study baseline and intervention phases.

## Primary outcome 2

### Identification of SARS-CoV-2 nosocomial transmission using sequencing data

For each HOCI case during the intervention phases, the occurrence of this outcome was defined as positive where the following two answers had been recorded in the Hospital Transmission section of the relevant clinical reporting form (CRF04):

> "Is sequencing report suggestive that patient is part of a hospital outbreak (i.e. involving ≥2 patients or HCWs in the hospital)?: Yes"
> &
> "If yes, was linkage to one or more of these patients suspected at initial IPC investigation?: No"

The occurrence of this outcome was considered to be negative if the following answer was recorded:

> "Is sequencing report suggestive that patient is part of a hospital outbreak (i.e. involving ≥2 patients or HCWs in the hospital)?: No"

Or if the following combination was recorded:

> "Is sequencing report suggestive that patient is part of a hospital outbreak (i.e. involving ≥2 patients or HCWs in the hospital)?: Yes"
> &
> "If yes, was linkage to one or more of these patients suspected at initial IPC investigation?: Yes"

The outcome will be considered missing if either the first question was not answered, or if the first question was answered 'Yes' and the second question was not answered or was answered 'unknown'.

The outcome was also considered negative if the viral sequence and sequence report had not been returned during the period of study data collection.

This outcome was only evaluated for study sequencing intervention periods.

## Secondary outcome 1

### Incidence of IPC-defined SARS-CoV-2 hospital outbreaks

An IPC-defined SARS-CoV-2 hospital outbreak was defined as the occurrence of at least two HOCI cases on the same ward, with at least one having an interval of ≥8 days from admission to symptom onset (if known) or sample date, and with the outbreak event considered to be concluded if there was a gap of 28 days before the observation of another HOCI case (*Public Health England, 2020*). This was evaluated using the ward location recorded at patient registration into the study (CRF01) for HOCI cases, cross-checked against patient movement data to confirm location at diagnostic sampling. Outbreak events were considered to have occurred on the date of diagnosis of the first HOCI case. This outcome was evaluated for study baseline and intervention phases.

## Secondary outcome 2

### Changes to IPC actions following receipt of sequencing report

For each HOCI case, the occurrence of this outcome was defined as positive if 'Yes' is the answer to either of the following two questions in the 'Sequencing report impact on IPC team' section of CRF04:

> "Overall, did the sequencing report change IPC practice for this ward?"
> And/or
> "Has the sequencing report information been used in IPC decisions beyond this patient's ward?"

And/or, if any specific changes to IPC practice were recorded on CRF04.

The occurrence of the outcome was considered negative if at least one of these questions was answered 'No' and neither is answered 'Yes', and it was considered missing if neither were answered.

This outcome was only evaluated for study sequencing intervention periods.

### Secondary outcome 3

Ideal changes to IPC actions following receipt of sequencing report
A binary outcome was defined for each HOCI patient. This was based on the value of secondary outcome 3, but was additionally defined as positive (whether secondary outcome 3 was negative or missing) if an 'increase' or 'decrease' that was not implemented was recorded for any of the actions in the 'other recommended changes to IPC protocols' section of CRF04.

This outcome was only evaluated for study sequencing intervention periods.

### Secondary outcome 4

Incidence of IPC + sequencing-defined SARS-CoV-2 hospital outbreaks
An IPC + sequencing-defined SARS-CoV-2 hospital outbreak was defined as the occurrence of at least two HOCI cases on the same ward that form a genetic cluster with maximum viral sequence pairwise SNP distance of 2 between each individual included and their nearest neighbour within the cluster. This was evaluated using the ward location recorded at patient registration into the study (CRF01), with HOCI cases sorted into outbreak groups using the lists of close sequence matches on unit-ward as returned by the SRT and recorded in CRF03.

Outbreak events were considered to have occurred on the date of diagnosis of the first HOCI case. This outcome was evaluated for study sequencing intervention periods for all sites.

### Secondary outcome 5

HCW sickness
The proportion of HCWs on sick leave due to COVID-19 was calculated using the 'Current staffing levels on ward' section of CRF02. Analysis was performed using the first available data within each IPC-defined SARS-CoV-2 hospital outbreak (as per secondary outcome 1), so as to provide a measure of the level of staff absence at the start of each outbreak. This outcome was evaluated for study baseline and intervention phases.

### Changes with respect to the SAP

It was planned that the cumulative proportion of HCWs vaccinated at each site for each study week would also be included as a covariate for the analysis models of incidence outcomes. However, this was dropped because these supplementary data could not be obtained from four sites (and one site was only able to provide partial data). Where available, the data showed the rollout of HCW vaccination to be broadly consistent across sites. As such, any effect of HCW vaccination on the incidence outcomes would be incorporated into estimates of variation in relation to calendar time.

Local community incidence was not included within the SAP, but was added as an adjustment factor for incidence models because within-hospital prevalence of patients admitted for SARS-CoV-2 did not correlate perfectly with local incidence (e.g. due to triage of COVID-19 patients to different hospitals).

The proportion of HOCI cases in which the sequencing report feedback was considered to be 'useful' was added as a secondary outcome.

It was stated in the SAP: "We will conduct sensitivity analyses excluding study sites and/or periods with suboptimal implementation of the trial intervention, both in terms of overall population sequencing coverage for HOCIs and the turnaround time for sequence reports being returned to IPC teams. The exact criteria for this will be decided amongst the study team before any analysis has been conducted." This forms the basis for the 'per protocol' analyses presented. It was not possible to prespecify the exact criteria. After data collection for the study had been completed, it became clear that none of the sites had met target turnaround times for sequence reporting in the intervention phases, and so it was decided to set criteria to select the 50% 'higher performing' sites.

It was stated in the SAP: "If the target turnaround time for sequence generation and reporting is missed for a substantial proportion of HOCI cases in each of the intervention phases, then results [for Impact on IPC actions] will also be reported separately for the subset of cases for which the intervention was implemented within the target timeframe." Because the proportion of HOCI cases with SRT report returned within the target time frame was low for the rapid intervention phases, we instead reported the association of this outcome with turnaround time more generally.

The SAP did not state that unadjusted estimates would be reported for the incidence rate ratios for HAIs and outbreaks, but these have been added for completeness in response to the comments of a Reviewer (*Appendix 1—table 5*).

Weekly incidence rates for outbreak events are displayed as '/1000 inpatients' rather than '/100 inpatients' to improve display.

## Small sample correction

The topic of small sample corrections for cluster randomised and other cluster-structured studies (e.g. stepped wedge trials) with outcomes that are not normally distributed is an area of ongoing active research. To our knowledge, there do not exist any studies regarding appropriate corrections for clustered data when analysing an outcome with negative binomial distribution. However, when calculating p-values and confidence intervals for the primary and secondary outcomes we will use a *t*-distribution with 12 or 13 degrees of freedom (*n* clusters – *n* relevant parameters) in order to ensure that there is not an inflated type 1 error rate. This correction has shown appropriate characteristics in simulation studies of analyses of binary outcomes using mixed effects models and generalised estimating equations (*Li and Redden, 2015*; *Thompson et al., 2021*).

## Decision regarding continuation of study into final phase

A decision regarding the final phase of the study (Period 4) was planned for April 2021, with the options being: ending of the study at Period 3, a further phase of rapid sequencing at each site or a further phase of 'baseline' data collection without use of the SRT. A recommendation regarding this decision was made by the study investigators and agreed with the TSC-DMC. The decision was determined by the course of the epidemic and the progress of vaccination among key risk groups, and by the quantity of data collected by the end of Period 3. The decision was not based on any interim evaluation of the effect of the sequencing intervention under investigation on the incidence of nosocomial infection.

A decision was made to stop the study at the end of Period 3 because the total sample size was close to that projected for the study, and few new HOCI cases were being recorded at this point in time.

## Further details for incidence model specification

The primary outcome of incidence of SARS-CoV-2 HAIs was analysed using a mixed effects negative binomial regression model. This acts as an extension of a Poisson regression model, with an additional parameter allowing for overdispersion (for a Poisson model the variance is always the mean for any given combination of covariables and conditional on random effect terms). The negative binomial regression model is a generalised linear model, which uses a log-link function for the expectation of the outcome variable. In our analysis, we used nested independent normally distributed random effects for each study site, and for each study phase within each study site, which were incorporated into the 'linear predictor' for the expectation of the outcome variable as for other forms of linear mixed effects model. The variance of the two random effects components of the model was estimated within the maximum likelihood estimation for the model.

The command used in Stata to run this model was of the form:

```
menbreg n_HAIs i.study_phase_analysis prop_cov_sp_* study_week_sp_*
community_inc_sp_*, exposure(exposure_pw) || site_anon: || study_phase:,
irr dispersion(constant)
```

Where the dataset under analysis included one row per study week at each site, with the variables defined as

> n_HAIs: Number of HAIs observed at each sites in each study week
> study_phase_analysis: Intervention phase for each study week, with separate categories for the first week of each intervention phase at each site (to allow for 1 week to pass before potential impact on incidence of HAIs)
> prop_cov_sp_*: Spline basis variables for proportion of inpatients who were SARS-CoV-2 positive at each site in each study week
> study_week_sp_*: Spline basis variables for adjustment for calendar time

community_inc_sp_*: Spline basis variables for local community incidence for each site in each study week

exposure_pw: person-weeks of SARS-CoV-2 negative inpatients at each site for each study week (i.e. sum of patient-time at-risk for nosocomial SARS-CoV-2 infection)

## Qualitative analyses

An exploratory, qualitative process evaluation using iterative programme theory employed semi-structured interviews with 39 diverse healthcare professionals between December 2020 and June 2021. Participants were purposively sampled from 5/14 sites. Data collection and analysis (deductive and inductive thematic analysis) focused on the programme theory: intervention acceptability; contextual dependencies; issues of fidelity/adaption; insights into local implementation; and effects on outcomes.

## Results

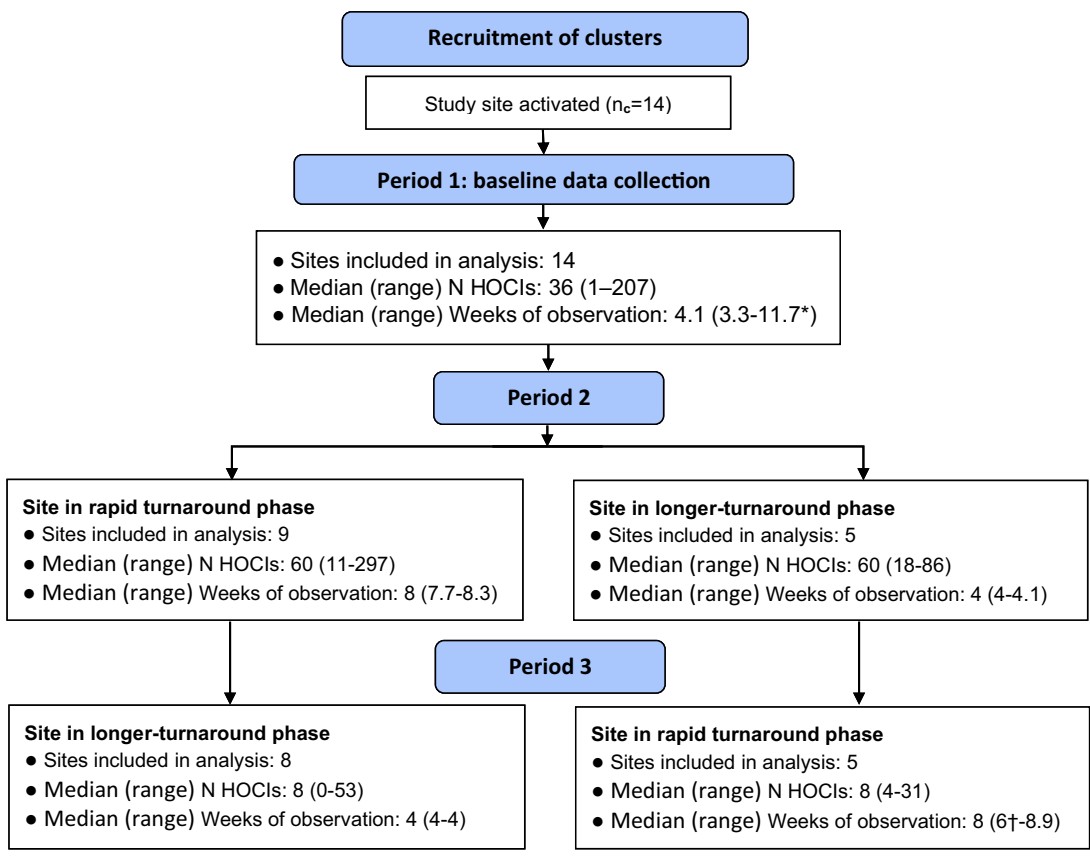

**Appendix 1—figure 1.** Flow diagram of study site enrolment and intervention implementation. *Baseline phase extended for one site due to a complete lack of hospital-onset COVID-19 infection (HOCI) cases during the first few weeks of study period and omission of longer-turnaround sequencing phase. †Rapid sequencing phase truncated at one site due to cessation of enrolment at all sites.

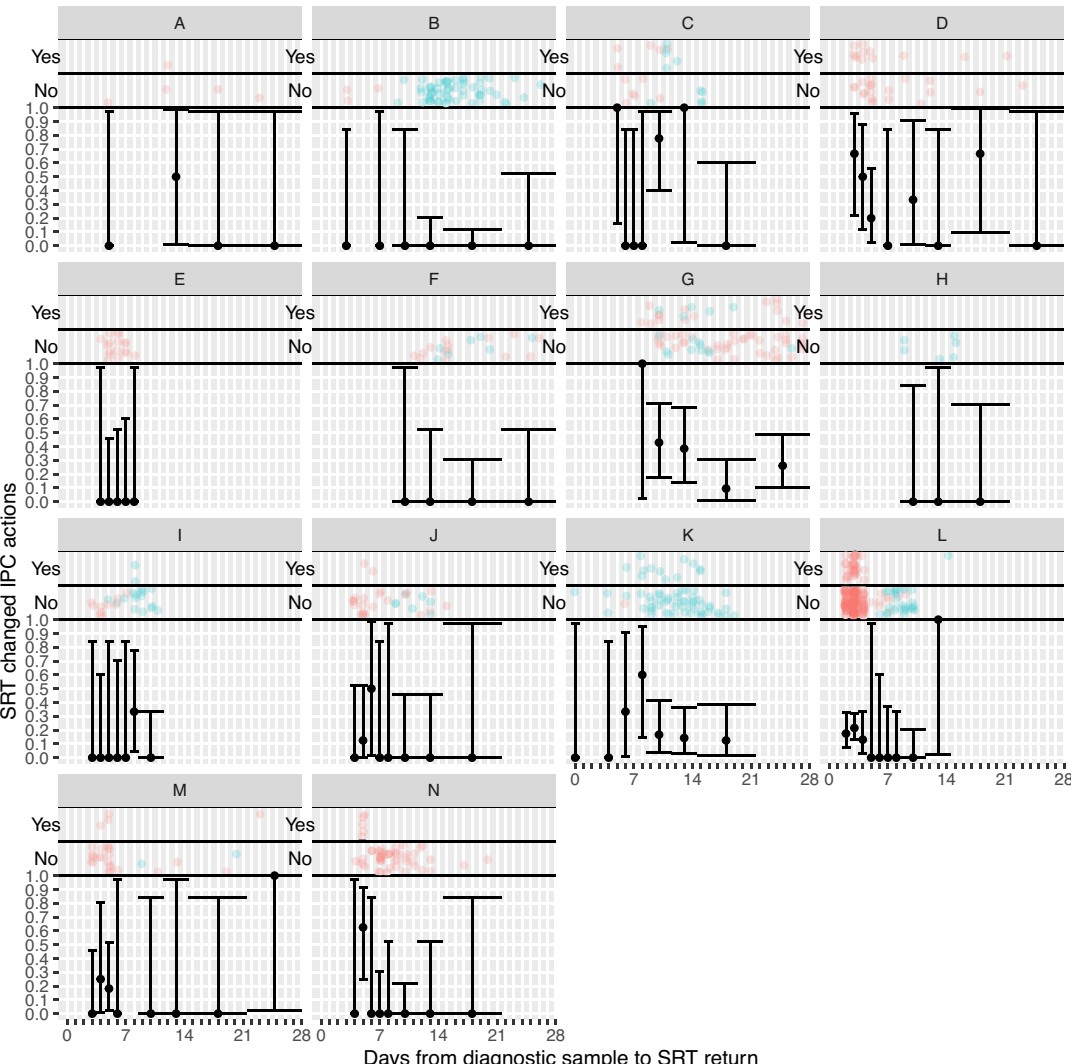

**Appendix 1—figure 2.** Plots of the proportion of returned sequence reporting tool (SRT) reports that had an impact on infection prevention and control (IPC) actions by study site. Results are only shown up to turnaround times of ≤28 days, and grouped proportions are shown for ≥9 days because of data sparsity at higher turnaround times. Error bars show binomial 95% CIs. 'Yes' and 'No' outcomes for individual hospital-onset COVID-19 infection (HOCI) cases are displayed, colour-coded by rapid (red) and longer-turnaround (blue) intervention phases and with random jitter to avoid overplotting.

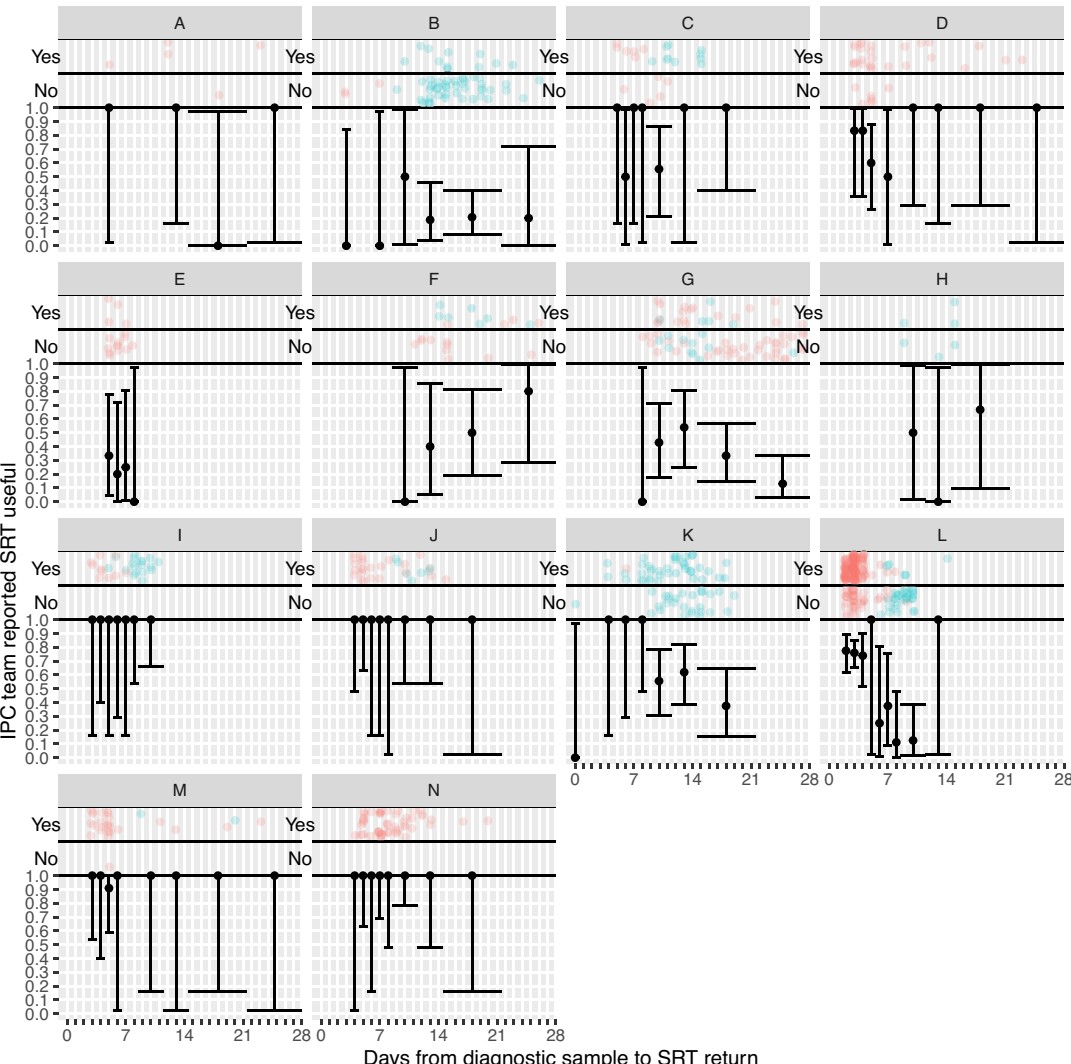

**Appendix 1—figure 3.** Plots of the proportion of returned sequence reporting tool (SRT) reports that were reported to be useful by infection prevention and control (IPC) teams by study site. Results are only shown up to turnaround times of ≤28 days, and grouped proportions are shown for ≥9 days because of data sparsity at higher turnaround times. Error bars show binomial 95% CIs. 'Yes' and 'No' outcomes for individual hospital-onset COVID-19 infection (HOCI) cases are displayed, colour-coded by rapid (red) and longer-turnaround (blue) intervention phases and with random jitter to avoid overplotting. 'Unsure' responses were coded as 'No'.

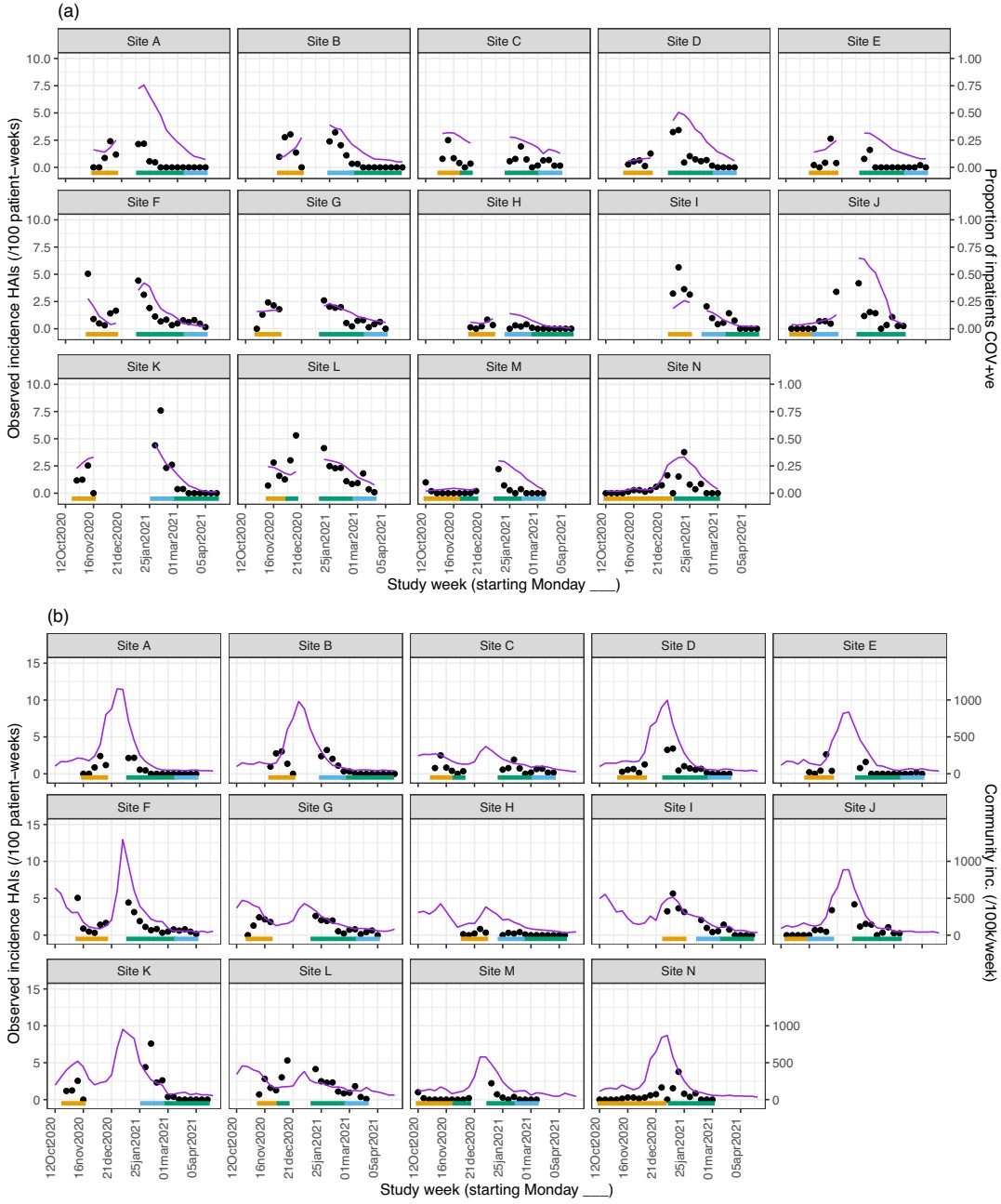

**Appendix 1—figure 4.** Weekly incidence of hospital-acquired infections (HAIs) at each site (●), with (**a**) proportion of all inpatients SARS-CoV-2+ve and (**b**) local community incidence of SARS-CoV-2+ve tests also plotted on the y-axis (purple line). Horizontal bars show the duration of study phases (orange: baseline; blue: longer turnaround; green: rapid).

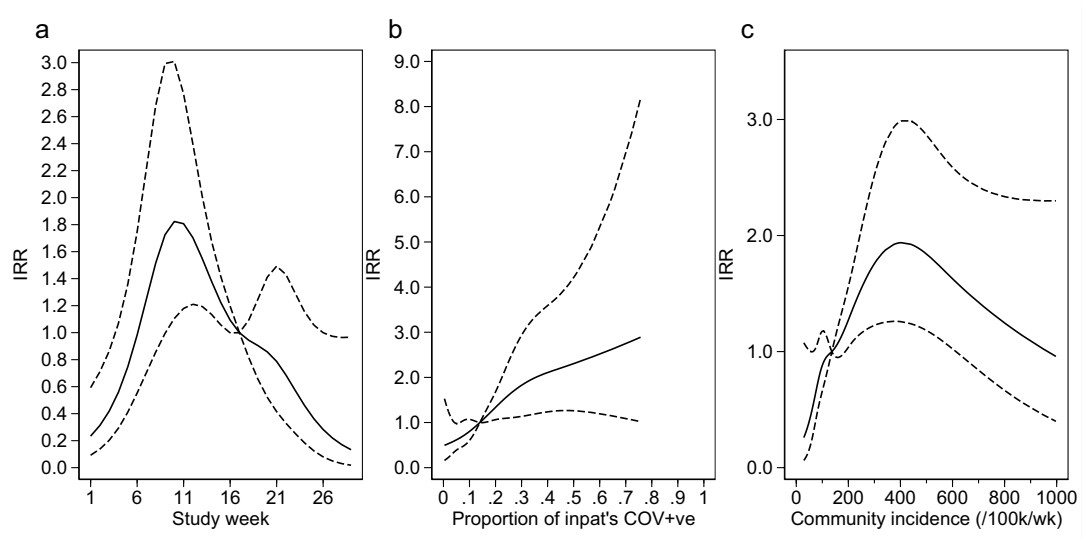

**Appendix 1—figure 5.** Adustment variables for analysis of weekly incidence of infection prevention and control (IPC)-defined hospital-acquired infections (HAIs) per 100 inpatients, as described in *Table 3*. Incidence rate ratios are displayed relative to the median for (**a**) calendar time expressed as study week from 12 October 2020, (**b**) proportion of inpatients with positive SARS-CoV-2 test, and (**c**) local community incidence of SARS-CoV-2 (government surveillance data weighted by total set of postcodes for patients at each site). The spline curves shown are estimated simultaneously within the final analysis model and show how these factors have independent contributions to the prediction of the incidence rate for HAIs. The associations for each covariable indicated by model parameter point estimates are shown as solid lines, with 95% CIs shown as dashed lines. Adjustment for (**c**) was not pre-specified in the statistical analysis plan (SAP), but adding this variable to the model was associated with a statistically significant improvement in fit (p=0.01). The proportion of community-sampled cases in the region that were found to be the Alpha variant on sequencing was also considered, but adding this as a linear predictor did not lead to a statistically significant improvement in model fit (p=0.78).

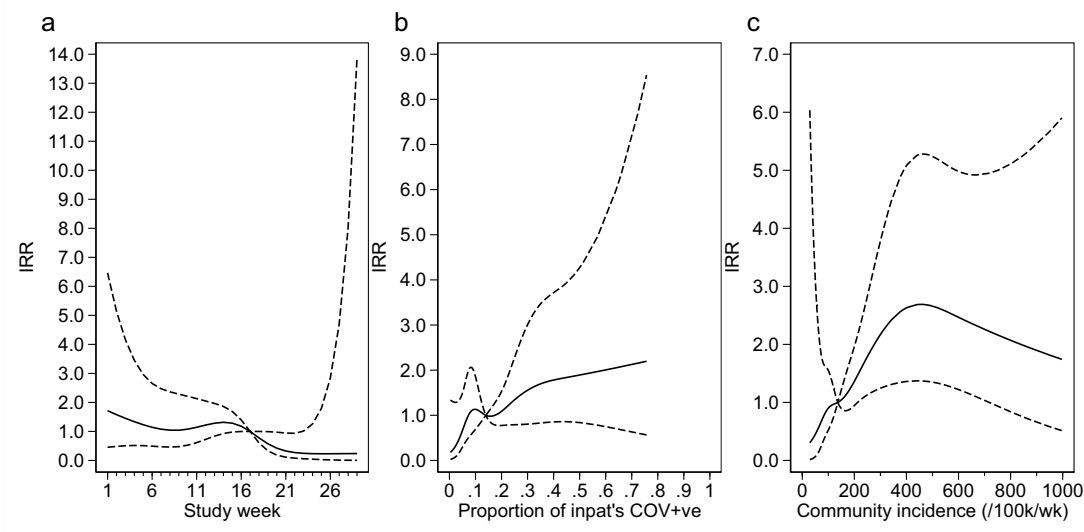

**Appendix 1—figure 6.** Adustment variables for analysis of weekly incidence of infection prevention and control (IPC)-defined outbreak events per 1000 inpatients, as described in *Table 3*. Incidence rate ratios are displayed relative to the median for (**a**) calendar time expressed as study week from 12 October 2020, (**b**) proportion of inpatients with positive SARS-CoV-2 test, and (**c**) local community incidence of SARS-CoV-2 (government surveillance data weighted by total set of postcodes for patients at each site). The spline curves shown are estimated simultaneously within the final analysis model and show how these factors have independent contributions to the prediction of the incidence rate for outbreaks. The associations for each covariable indicated
*Appendix 1—figure 6 continued on next page*

*Appendix 1—figure 6 continued*

by model parameter point estimates are shown as solid lines, with 95% CIs shown as dashed lines. Adjustment for (**c**) was not pre-specified in the statistical analysis plan (SAP), but adding this variable to the model was associated with a near-statistically significant improvement in fit (p=0.05) and was included for consistency with the analysis of individual hospital-acquired infections (HAIs). The proportion of community-sampled cases in the region that were found to be the Alpha variant on sequencing was also considered, but adding this as a linear predictor did not lead to a statistically significant improvement in model fit (p=0.80).

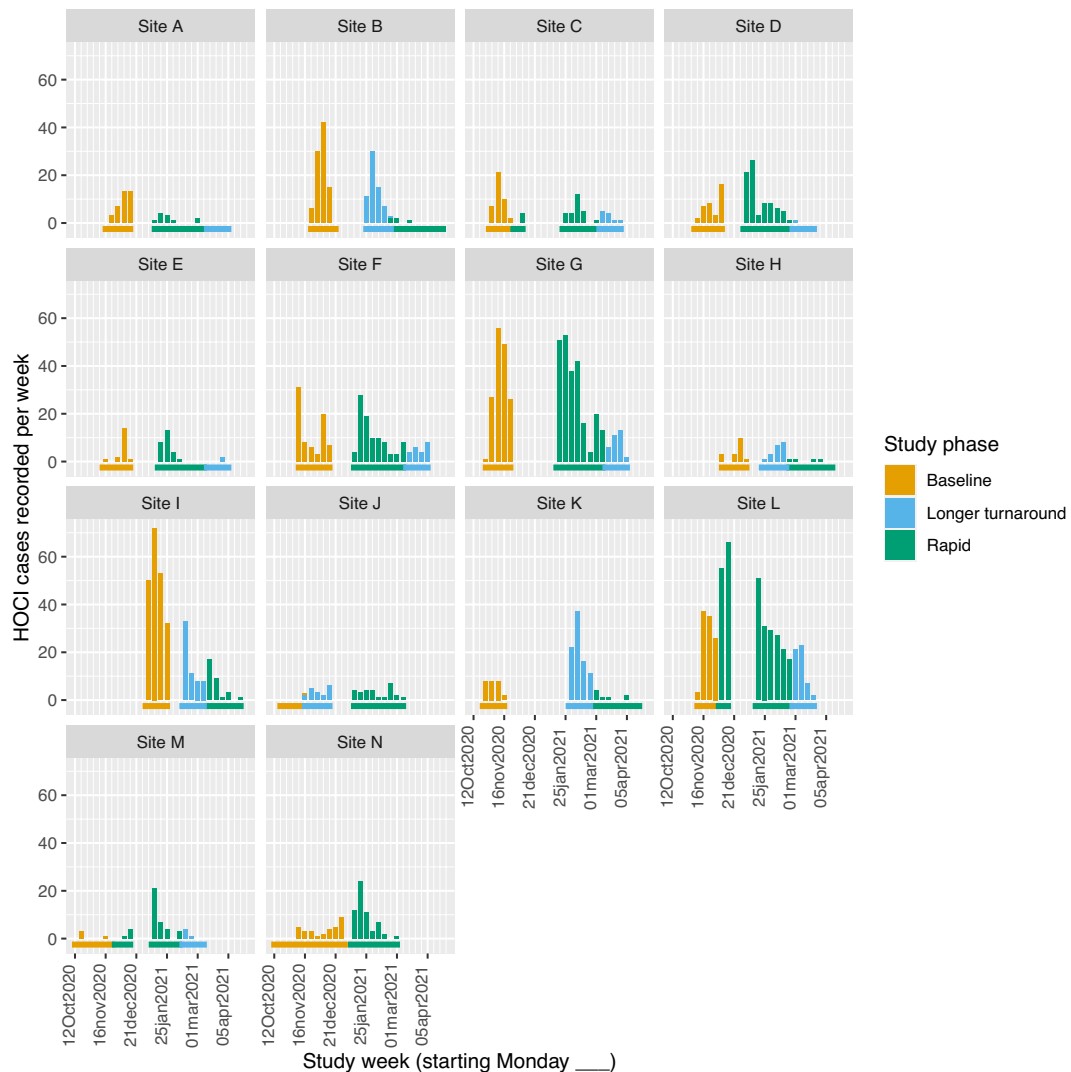

**Appendix 1—figure 7.** Weekly counts of enrolled hospital-onset COVID-19 infection (HOCI) cases by date of positive test return for each site, colour-coded by intervention phases. Horizontal bars show the duration of study phases.

**Appendix 1—table 1.** Per outbreak event outcomes by study intervention phase.

| | Study phase | | | |
| --- | --- | --- | --- | --- |
| | **Baseline** | **Longer-turnaround** | **Rapid** | **Total** |
| *IPC-defined outbreak events* | | | | |
| *n* outbreak events | 129 | 33 | 114 | 276 |
| n/N (%) of HOCI cases part of outbreak event | 682/850 (80.2) | 314/373 (84.2) | 763/947 (80.6) | 1759/2170 (81.1) |

*Appendix 1—table 1 Continued on next page*

*Appendix 1—table 1 Continued*

| | Study phase | | | |
| --- | --- | --- | --- | --- |
| | **Baseline** | **Longer-turnaround** | **Rapid** | **Total** |
| Number of HOCIs per outbreak event, median (IQR, range) | 5.0 (3–8, 2–43) | 5.0 (3–9, 2–24) | 4.0 (2–7, 2–31) | 4.0 (2–8, 2–43) |
| Proportion of HCWs on sick leave due to COVID-19, median (IQR, range) [n] | 0.13 (0.00–0.35, 0.00–0.50) [13] | 0.05 (0.00–0.18, 0.00–0.30) [7] | 0.20 (0.08–0.33, 0.00–0.89) [14] | 0.13 (0.00–0.31, 0.00–0.89) [34] |
| *IPC + sequencing-defined outbreak events* | | | | |
| *n* outbreak events | – | 41 | 135 | 176 |
| *n/N* (%) of HOCI cases part of outbreak event | – | 292/373 (78.3) | 705/947 (74.4) | 997/1320 (75.5) |
| Number of HOCIs per outbreak event, median (IQR, range) | – | 5.0 (2-8, 2-23) | 3.0 (2-6, 2-29) | 3.0 (2-7, 2-29) |
| *For first HOCI in outbreak:* | | | | |
| SRT changed IPC practice, *n/N* (%, 95% CI) | – | 4/41 (10.4, 0–21.0) | 19/133 (14.9, 6.6–23.2) | 23/174 (13.2) |
| SRT changed IPC practice for ward, *n/N* (%) | – | 2/35 (5.7) | 6/82 (7.3) | 8/117 (6.8) |
| SRT used in IPC decisions beyond ward, *n/N* (%) | – | 2/35 (5.7) | 10/82 (12.2) | 12/117 (10.3) |
| *IPC team reported SRT to be useful, n/N (%)* | | | | |
| Yes | – | 20/35 (57.1) | 51/82 (62.2) | 71/117 (60.7) |
| No | – | 9/35 (25.7) | 15/82 (18.3) | 24/117 (20.5) |
| Unsure | – | 6/35 (17.1) | 16/82 (19.5) | 22/117 (18.8) |
| SRT would ideally have changed IPC practice, *n/N* (%*, 95% CI) | – | 4/41 (9.8) | 19/133 (14.3) | 23/174 (13.2) |

Odds ratio of SRT changed IPC practice for 'rapid vs. longer-turnaround' phases 1.54 (95% CI 0.37–6.44; p=0.52).

HCW, healthcare worker; HOCI, hospital-onset COVID-19 infection; IPC, infection prevention and control; IQR, interquartile range; SRT, sequence reporting tool.

*Estimated marginal value from mixed effects model, not raw %, evaluated on intention-to-treat basis with lack of SRT report classified as 'No'.

**Appendix 1—table 2.** Descriptive summary of impact of sequencing on IPC actions implemented during study intervention phases, as recorded on pre-specified study reporting forms.

| | Study phase | | | | | |
| --- | --- | --- | --- | --- | --- | --- |
| | **Longer-turnaround sequencing** | | | **Rapid turnaround sequencing** | | |
| *N* HOCI cases | 373 | | | 947 | | |
| *Review of IPC actions already taken* | *Support* | *Refute* | *Missing* | *Support* | *Refute* | *Missing* |
| SRT results support or refute IPC actions already taken* | 200/213 (93.9) | 7/213 (3.3) | 2 | 389/428 (90.9) | 9/428 (2.1) | 7 |
| *Changes to IPC practice following SRT* | *To enhanced* | *To routine* | *No change* | *To enhanced* | *To routine* | *No change* |
| Change to cleaning protocols on ward | 2/185 (1.1) | 0/185 (0.0) | 183/185 (98.9) | 7/341 (2.1) | 0/341 (0.0) | 334/341 (97.9) |
| | *To greater* | *To fewer* | *No change* | *To greater* | *To fewer* | *No change* |
| Change to visitor restrictions | 1/186 (0.5) | 0/186 (0.0) | 185/186 (99.5) | 1/340 (0.3) | 0/340 (0.0) | 339/340 (99.7) |
| | *To 'cohort nursing'* | *To 'other restrictions'* | *No change* | *To 'cohort nursing'* | *To 'other restrictions'* | *No change* |
| Change to staffing restrictions on ward | 0/186 (0.0) | 1/186 (0.5) | 185/186 (99.5) | 0/336 (0.0) | 1/336 (0.3) | 335/336 (99.7) |
| | *Increase* | *Decrease* | *No change* | *Increase* | *Decrease* | *No change* |
| Hand hygiene audit frequency | 1/185 (0.5) | 0/185 (0.0) | 184/185 (99.5) | 10/335 (3.0) | 0/335 (0.0) | 325/335 (97.0) |
| IPC staff visits to ward | 1/185 (0.5) | 0/185 (0.0) | 184/185 (99.5) | 14/335 (4.2) | 0/335 (0.0) | 321/335 (95.8) |
| Assessment of alcogel stocks | 0/185 (0.0) | 0/185 (0.0) | 185/185 (100.0) | 2/335 (0.6) | 0/335 (0.0) | 333/335 (99.4) |
| Assessment of soap stocks | 0/185 (0.0) | 0/185 (0.0) | 185/185 (100.0) | 2/334 (0.6) | 0/334 (0.0) | 332/334 (99.4) |

*Appendix 1—table 2 Continued on next page*

*Appendix 1—table 2 Continued*

| | Study phase | | | | | |
|---|---|---|---|---|---|---|
| | **Longer-turnaround sequencing** | | | **Rapid turnaround sequencing** | | |
| Assessment of aseptic non-touch technique compliance | 0/185 (0.0) | 0/185 (0.0) | 185/185 (100.0) | 8/335 (2.4) | 0/335 (0.0) | 327/335 (97.6) |
| Assessment of PPE supply | 1/185 (0.5) | 0/185 (0.0) | 184/185 (99.5) | 9/336 (2.7) | 0/336 (0.0) | 327/336 (97.3) |
| Availability of doffing and donning buddy | 0/185 (0.0) | 0/185 (0.0) | 185/185 (100.0) | 1/333 (0.3) | 0/333 (0.0) | 332/333 (99.7) |
| IPC signage assessment | 1/185 (0.5) | 0/185 (0.0) | 184/185 (99.5) | 12/336 (3.6) | 0/336 (0.0) | 324/336 (96.4) |
| IPC signage implementation | 1/185 (0.5) | 0/185 (0.0) | 184/185 (99.5) | 11/336 (3.3) | 0/336 (0.0) | 325/336 (96.7) |
| Training on IPC procedures | 0/185 (0.0) | 0/185 (0.0) | 185/185 (100.0) | 8/336 (2.4) | 0/336 (0.0) | 328/336 (97.6) |

Data shown as *n* or *n/N* (%). Overall impact on IPC actions per HOCI case is given in **Table 2**.
HOCI, hospital-onset COVID-19 infection; IPC, infection prevention and control; PPE, personal protective equipment; SRT, sequence reporting tool.
*Sites could select 'Yes' or 'No' for both 'Support' and 'Refute' as these were entered as separate data items.

**Appendix 1—table 3.** Descriptive summary of impact of sequencing on IPC actions implemented during study intervention phases, only including the first HOCI in each IPC+sequencing-defined outbreak event, as recorded on pre-specified study reporting forms.

| | Study phase | | | | | |
|---|---|---|---|---|---|---|
| | **Longer-turnaround sequencing** | | | **Rapid turnaround sequencing** | | |
| N HOCI cases | 41 | | | 135 | | |
| *Review of IPC actions already taken* | *Support* | *Refute* | *Missing* | *Support* | *Refute* | *Missing* |
| SRT results support or refute IPC actions already taken* | 30/35 (85.7) | 3/35 (8.6) | 0 | 71/82 (86.6) | 5/82 (6.1) | 2 |
| *Changes to IPC practice following SRT* | *To enhanced* | *To routine* | *No change* | *To enhanced* | *To routine* | *No change* |
| Change to cleaning protocols on ward | 1/34 (2.9) | 0/34 (0.0) | 33/34 (97.1) | 1/70 (1.4) | 0/70 (0.0) | 69/70 (98.6) |
| | *To greater* | *To fewer* | *No change* | *To greater* | *To fewer* | *No change* |
| Change to visitor restrictions | 0/34 (0.0) | 0/34 (0.0) | 34/34 (100.0) | 1/70 (1.4) | 0/70 (0.0) | 69/70 (98.6) |
| | *To 'cohort nursing'* | *To 'other restrictions'* | *No change* | *To 'cohort nursing'* | *To 'other restrictions'* | *No change* |
| Change to staffing restrictions on ward | 0/34 (0.0) | 1/34 (2.9) | 33/34 (97.1) | 0/70 (0.0) | 1/70 (1.4) | 69/70 (98.6) |
| | *Increase* | *Decrease* | *No change* | *Increase* | *Decrease* | *No change* |
| Hand hygiene audit frequency | 0/34 (0.0) | 0/34 (0.0) | 34/34 (100.0) | 3/70 (4.3) | 0/70 (0.0) | 67/70 (95.7) |
| IPC staff visits to ward | 0/34 (0.0) | 0/34 (0.0) | 34/34 (100.0) | 5/70 (7.1) | 0/70 (0.0) | 65/70 (92.9) |
| Assessment of alcogel stocks | 0/34 (0.0) | 0/34 (0.0) | 34/34 (100.0) | 2/70 (2.9) | 0/70 (0.0) | 68/70 (97.1) |
| Assessment of soap stocks | 0/34 (0.0) | 0/34 (0.0) | 34/34 (100.0) | 2/70 (2.9) | 0/70 (0.0) | 68/70 (97.1) |
| Assessment of aseptic non-touch technique compliance | 0/34 (0.0) | 0/34 (0.0) | 34/34 (100.0) | 3/70 (4.3) | 0/70 (0.0) | 67/70 (95.7) |
| Assessment of PPE supply | 0/34 (0.0) | 0/34 (0.0) | 34/34 (100.0) | 3/70 (4.3) | 0/70 (0.0) | 67/70 (95.7) |
| Availability of doffing and donning buddy | 0/34 (0.0) | 0/34 (0.0) | 34/34 (100.0) | 1/70 (1.4) | 0/70 (0.0) | 69/70 (98.6) |
| IPC signage assessment | 0/34 (0.0) | 0/34 (0.0) | 34/34 (100.0) | 4/70 (5.7) | 0/70 (0.0) | 66/70 (94.3) |
| IPC signage implementation | 0/34 (0.0) | 0/34 (0.0) | 34/34 (100.0) | 4/70 (5.7) | 0/70 (0.0) | 66/70 (94.3) |
| Training on IPC procedures | 0/34 (0.0) | 0/34 (0.0) | 34/34 (100.0) | 4/70 (5.7) | 0/70 (0.0) | 66/70 (94.3) |

Data shown as *n* or *n/N* (%). Overall impact on IPC actions per HOCI case is given in **Appendix 1—table 1**.
HOCI, hospital-onset COVID-19 infection; IPC, infection prevention and control; PPE, personal protective equipment; SRT, sequence reporting tool.
*Sites could select 'Yes' or 'No' for both 'Support' and 'Refute' as these were entered as separate data items.

**Appendix 1—table 4.** Per-sample costs of SARS-CoV-2 genome rapid and longer-turnaround sequencing.

| Laboratories | Lab 1 | Lab 2 | Lab 3 | Lab 4 | Lab 5 | Lab 6 | Lab 7 | Lab 8 | Lab 9 | Lab 10 |
|---|---|---|---|---|---|---|---|---|---|---|
| Rapid turnaround sequencing | | | | | | | | | | |

*Appendix 1—table 4 Continued on next page*

*Appendix 1—table 4 Continued*

| Laboratories | Lab 1 | Lab 2 | Lab 3 | Lab 4 | Lab 5 | Lab 6 | Lab 7 | Lab 8 | Lab 9 | Lab 10 | |
|---|---|---|---|---|---|---|---|---|---|---|---|
| Sequencing platform | Illumina MiSeq | Nanopore MinION/ GridiON | Nanopore GridiON | Nanopore GridiON | Nanopore GridiON | Nanopore MinION/ GridiON | Nanopore GridiON | Nanopore GridiON | Illumina MiSeq | Illumina MiSeq | Mean |
| Batch size | 24 | 24 | 24 | 96 | 24 | 24 | 24 | 24 | 96 | 96 | |
| Equipment | £45.11 | £26.06 | £19.34 | £4.38 | £12.38 | £24.66 | £11.99 | £11.26 | £5.91 | £6.13 | £16.72 |
| Consumables | £69.14 | £54.56 | £87.07 | £31.11 | £79.06 | £28.84 | £62.09 | £46.02 | £14.37 | £39.63 | £51.19 |
| Staff | £6.11 | £20.25 | £24.66 | £7.93 | £11.16 | £5.66 | £12.16 | £8.45 | £2.20 | £3.45 | £10.20 |
| Total per-sample cost | £120.36 | £100.87 | £131.07 | £43.43 | £102.60 | £59.17 | £86.23 | £65.73 | £22.48 | £49.21 | £78.11 |
| Total cost (including overheads calculated at 20%) | £144.43 | £121.04 | £157.28 | £52.11 | £123.12 | £71.01 | £103.48 | £78.88 | £26.97 | £59.05 | £93.74 |

**Longer-turnaround sequencing**

| Sequencing platform | Illumina MiSeq | Nanopore MinION/ GridiON | Nanopore GridiON | Nanopore GridiON | Nanopore GridiON | Nanopore MinION/ GridiON | Nanopore GridiON | Nanopore GridiON | Nanopore MinION | Illumina MiSeq | Mean |
|---|---|---|---|---|---|---|---|---|---|---|---|
| Batch size | 24 | 24 | 24 | 96 | 24 | 24 | 24 | 96 | 24 | 96 | |
| Equipment | £40.60 | £22.15 | £17.02 | £3.94 | £11.88 | £22.44 | £11.27 | £2.81 | £2.54 | £5.76 | £14.04 |
| Consumables | £61.53 | £48.56 | £77.49 | £27.69 | £70.36 | £25.67 | £55.26 | £11.51 | £33.75 | £35.27 | £44.71 |
| Staff | £4.95 | £15.19 | £16.52 | £2.78 | £2.23 | £4.53 | £12.04 | £8.45 | £11.85 | £3.32 | £8.19 |
| Total per-sample cost | £107.08 | £85.89 | £111.03 | £34.41 | £84.48 | £52.65 | £78.56 | £22.77 | £48.13 | £44.34 | £66.94 |
| Total cost (including overheads calculated at 20%) | £128.50 | £103.07 | £133.24 | £41.29 | £101.38 | £63.18 | £94.28 | £27.33 | £57.76 | £53.21 | £80.32 |

**Appendix 1—table 5.** Incidence outcomes by study intervention phase with unadjusted IRR.

| | Study phase | | | IRR[†] (95% CI, p-value) | |
|---|---|---|---|---|---|
| | **Baseline** | **Longer-turnaround** | **Rapid** | **Longer-turnaround vs. baseline** | **Rapid vs. baseline** |
| *All sites* | | | | | |
| *n* HOCI cases | 850 | 373 | 947 | – | – |
| *n* IPC-defined HAIs | 488 | 207 | 576 | – | – |
| Weekly incidence of IPC-defined HAIs per 100 inpatients, mean (median, IQR, range)* [primary outcome] | 1.0 (0.5, 0.0–1.4, 0.0–5.6) | 0.7 (0.3, 0.0–0.7, 0.0–7.6) ‡ | 0.6 (0.3, 0.0–0.8, 0.0–5.3) ‡ | 0.49 (0.21–1.19, 0.12) | 0.47 (0.21–1.08, 0.07) |
| *n* IPC-defined outbreak events | 129 | 33 | 114 | – | – |
| Weekly incidence of IPC-defined outbreak events per 100 inpatients, mean (median, IQR, range)* | 0.3 (0.1, 0.0–0.4, 0.0–2.3) | 0.1 (0.0, 0.0–0.1, 0.0–0.9) ‡ | 0.1 (0.0, 0.0–0.0, 0.0–0.9) ‡ | 0.25 (0.10–0.66, 0.005) | 0.23 (0.10–0.54, 0.001) |

IPC-defined HAIs are considered to be 'probable' or 'definite' HAIs.

HAI, hospital-acquired infection; HOCI, hospital-onset COVID-19 infection; IPC, infection prevention and control; IQR, interquartile range; IRR, incidence rate ratio.

*Descriptive data over all week-long periods at all study sites.

[†]Without adjustment for proportion of current inpatients at site that are COVID-19 cases, community incidence rate, and calendar time.

‡Not including data from the first week of each intervention period, or in the week following any break in the intervention period.

