## [Editor Report]

This article contains valuable information on the potential value of real-time genome sequencing to inform infection control practices. The study, unique in its size, addresses the implementation of this approach during the height of the COVID-19 pandemic. Naturally, the extreme situation limited the options for choices in infection control practices.

---

## [Decision Letter]

**Decision letter after peer review:**

Thank you for submitting your article "Effectiveness of rapid SARS-CoV-2 genome sequencing in supporting infection control for hospital-onset COVID-19 infection: multicenter, prospective study" for consideration by *eLife*. Your article has been reviewed by 3 peer reviewers, including Marc J Bonten as Reviewing Editor and Reviewer #1, and the evaluation has been overseen by Jos van der Meer as the Senior Editor. The following individuals involved in review of your submission have agreed to reveal their identity: Ben S Cooper and Mark Pritchard (Reviewer #3).

The questions raised by the reviewers address clarifications and interpretation of data, and we invite you to respond to these.

*Reviewer #1 (Recommendations for the authors):*

Interpretational questions, comments

The turnaround times achieved are, definitely for the "rapid" part, much longer than was targeted, with overall 9,3% reported in time. This implies that even in the settings tested (which I consider "as good as it gets") these targets were too optimistic. Yet, there is a clear difference in reporting time between the 2 strategies, and that difference, as achieved, was not associated with a beneficial outcome on any of the measured outcome parameters. In fact, the proportion of HOCI in which an unexpected linkage was detected based upon WGS was low in both study phases. And the changes in IPC that were realized were infrequent (7.4-7.8%), and associated with short turnaround times. My interpretation would be that reporting of WGS was still too slow to act as a true intervention, and, therefore, no effects could be demonstrated, as stated "In 91.9%, (589/641, Table S2) of cases the reports were interpreted as supportive of IPC actions already taken."

Nevertheless, it was stated that "Extensive qualitative analyses[15] found high levels of acceptability for the SRT sequencing reports, which supported decision-making about IPC activity (e.g. stand down some IPC actions or continue as planned)."

I think that the authors are overly optimistic about the impact of WGS reporting in this study in the discussion:

Line 295: "Our study provides a substantial body of evidence regarding the introduction of viral WGS into hospital functioning, routine IPC practice, its potential impact on outbreak management and the challenges that need to be overcome to achieve implementation across the UK." I don't understand what is meant with "that the study provides evidence for its potential impact", as statistically significant impact is not demonstrated for any of the study outcomes.

Line 346: "Our study provides the first evidence that with faster turnaround times, viral sequences can inform ongoing IPC actions in managing nosocomial infections; results returned within {less than or equal to}5 days from sampling to result changed the actions of IPC teams in around 20% of cases." On an intention-to-treat basis there was no significant effect on the occurrence of HOCI, and it doesn't become clear from Tables S2 and S3 that IPC practices were changed importantly. In fact, according to these tables it looks as if hardly any IPC actions were changed. It is not clear from these tables if – in instances where measures were taken – more measures were taken simultaneously. That could be made more clear. For instance in Table S3 for Rapid and support: were measures taken in 5 of 70 outbreaks or in 30 of 70 (if you sum up all individual measures).

The research question remains whether even shorter turnaround times for more patients with HOCI could have an impact on transmissions and – ultimately – patient outcome. I would think that WGS could only have such an impact if IPC measures would be different for different genotypes or would be different if transmission was demonstrated. The authors could better explain why and under what circumstances they expect that WGS reporting could impact patient care.

*Reviewer #2 (Recommendations for the authors):*

1. The study appears to have excluded cases where linkage between cases was established on epidemiological grounds before sequencing occurred. This both makes it more difficult to see an effect of sequencing and begs the question about the intensity and comparability of IPC programs across the study hospitals. Hospitals with aggressive and capable IPC teams may have very good insight into nosocomial case sources and factors by virtue of their epidemiologic evaluations and thus have less room to benefit from sequencing.

2. The investigators report that nosocomial infection was not suspected in ~7% of HOCI cases. Please contextualize by stating what % of HOCI cases were suspected to be nosocomial.

3. It would be helpful to describe in the main body of the manuscript the kinds of actions that were expected of IPC programs and how they might decrease hospital-acquired infections.

4. Please clarify the criteria for higher performer sites. On the face of it, SRT in ~40% of cases and median time from diagnostic sample to SRT of up to 8 days doesn't seem like high performance relative to the study goals. Were there other aspects to these sites that differentiated them besides sequencing speed? Did they have larger or more experienced or more aggressive infection control teams just as they appear to have more effective sequencing teams?

5. Was there a standardized protocol for how IPC teams were to respond to sequencing data? Without this it's possible that opportunities existed to reduce transmission based on sequencing reports that were not taken.

6. Did study sequencing and SRTs incorporate healthcare worker test results? If so, please describe testing requirements for healthcare workers. Did they get serial testing or only in response to symptoms or exposures? If cluster detection did not include healthcare worker test results then this could lead to substantial underdetection of nosocomial events as well as obscure to IPCs the source of nosocomial cases and thus how best to contain them.

7. Please comment on the median incubation period for hospital-onset cases. With α and subsequent variants, incubation periods have become quite short thus progressively increasing the importance of speed in sequencing to have a potential impact on infections.

8. In our hospital experience, the highest value from sequencing has not been control of live outbreaks but rather insights gained regarding the robustness of global infection control policies (is a surgical mask protective, do AGPs increase risk, what is the risk of transmission between people in shared rooms, etc.). Hospitals that choose to act on these insights will make structural changes to their IPC programs (e.g. switch from surgical mask to respirators, change testing practices around AGPs, decrease use of shared rooms, increase ventilation within rooms, etc.) that ideally will not just be cluster-specific but lasting changes. This means that the incremental value of sequencing may decrease over time to the extent that one puts these successive changes into place. But sequencing itself was invaluable in providing the insights that led to these changes. In addition, these insights and policy changes are arguably less sensitive to turnaround time – the insights themselves are key but the responses transcend any one cluster and thus are less time sensitive.

*Reviewer #3 (Recommendations for the authors):*

1. The potential effect of the intervention depends on infection prevention measures that would have been taken in the absence of sequencing, and how hospitals' infection prevention and control teams respond to the sequencing results. There is therefore considerable potential for variability in the effect of the intervention between hospitals. From the methods it is not clear how analyses accounted for the effect of differences between hospitals. More generally, there is a lack of transparency as to precisely what the modelling assumptions were. It is important to either specify the models used unambiguosly using appropriate notation (including details on link functions and all random effects) in the methods or to provide the full STATA code used for the analysis (ideally, of course, both of these things should be done). It also important to specify which analyses were pre-specified in a statistical analysis plan and which were unplanned.

2. What the authors refer to as the "per protocol" sensitivity analysis in fact seems to be an unplanned sub-group analysis. Given that none of the trusts completely adhered to the protocol referring to this as a "per protocol" analysis seems misleading.

3. It is also important to see the difference between hospitals displayed in the results. Taking Figure 2 as an example, as the results are currently presented it is impossible to tell whether trends with increased time to result represent trends within each hospital, or a difference between those hospitals that tend to produce rapid results compared to those that tend to produce less rapid results. It should be a simple matter to produce a graph showing unaggregated results from each hospital together with the fitted model (and prediction intervals).

4. As the authors discuss, this trial took place during a period of significant variation in covid-19 prevalence, both in the community and among hospital patients. They report having adjusted their results for this. It would be useful to see both unadjusted and adjusted results to show the impact of this adjustment.

5. One hospital did not participate in the longer turnaround arm as this was considered inferior to their standard practice. Please describe current levels of whole genome use in these hospitals so that it is explicit how the intervention differs from the baseline conditions.

6. Please re-check the odds ratio on line 222. The rapid sequencing has a greater proportion of finding it helpful so the odds ratio should be greater than 1. If this is correct due to adjustment, please report this as adjusted OR and add details of adjustment.

7. It is important that the discussion starts with a clear statement that for the two primary outcomes no benefit of either intervention was shown and that this was true both in the main analysis and in the sensitivity analysis.

8. Line 274 "Among sites with the most effective implementation of the sequencing intervention we showed that feedback within 5 days of diagnosis allowed for maximal impact on IPC actions". This appears to be an unplanned subgroup analysis of a secondary outcome, and as such needs to come with suitable caveats.

9. Line 317 " may reflect a shorter incubation time for the Α variant compared to earlier variants" While there has been speculation about shorter incubation times for the Α variant, we are not aware of strong evidence that supports this, and recent analysis suggests similar generation time for the Α variant and earlier variants (see, for example, https://doi.org/10.1101/2021.12.08.21267454 which has now been accepted for publication in *eLife*).

10. Line 345. "Our study provides the first evidence that with faster turnaround

times, viral sequences can inform ongoing IPC actions in managing nosocomial infections; " While generally the paper strikes an appropriate note of cautious optimism for the approach despite the negative results in this trial, and it is understandable that the authors might want to emphasize the more positive aspects of the results when considering implications for sequencing to inform IPC measures, there is a danger that this statement (and the whole paragraph) is interpreted as attempting to put a positive spin on an essentially negative result. Thus, while it is reasonable to mention that in a minority of cases IPC actions were informed by viral sequencing data, it is equally important to emphasize that there was no evidence those measures had a positive impact on patient outcomes (it is surely equally possible that IPC actions informed by sequencing had a negative impact). Furthermore, it needs to be emphasized that in the main (planned) analysis there was no difference in the number of IPC actions informed by the sequencing results in rapid and longer-turnaround arms. It would be helpful to add here an explicit acknowledgement of the uncertain benefits and resource implications in this paragraph along with a discussion of how we might robustly evaluate the effectiveness (and cost-effectiveness) of WGS methods for IPC in the future, and in particular, emphasising what we learn from this trial.

11. Lines 596-602 Two planned secondary outcomes are not reported because they were either not thought to be useful or were insufficiently complete. While it seems reasonable to not conduct formal analysis for these reasons, surely these outcomes should still be reported somewhere given they were specified in the protocol and data does seem to have been obtained for them. Clearly, if researchers are permitted to selectively choose which of the planned outcomes to report there is large potential for bias.

Additional comments

1. line 528-9 "Site A observed 0 HOCI cases " – presumably this is just for longer-turnaround phase, as A appears on the graph in the rapid phase.

2. Figures S3 and S4: solid and dashed lines are not defined. Also, shading might be clearer than the dashed lines.

3. In Table 3, a number of outcomes are reported as 0.0 per 100 patients. It would be more helpful to display these in a unit that allows easy display of at least one significant figure, e.g. per 10 000.

4. Some figures use the colours red and green as the only way to distinguish between baseline and rapid periods. The additional distinction might help readers who are unable to distinguish between these colours.

5. The results have a subsection titled health economics. This is limited to laboratory costs, and has no reference to changes in outcomes or costs or savings associated with changing practice on the wards. We suggest changing the title to costs of tests.

6. Lab 9 appears to have had larger batch sizes and a significant cost saving associated with rapid testing. Is this due to the difference in prevalence at the different times they were following each protocol, or is there a difference in practice that could be applied more widely?

7. It is interesting to note how much more frequently teams reported finding the report useful compared to how often they reported using it to change practice. Was any information collected on this? Were they reporting reassurance from the results or were they useful in some other way?

8. Please define IRR on first use in the text and abstract.

9. Abstract and line 200 "We did not detect a statistically significant change".

Recommendations in the statistical literature (e.g. consensus statement from the ASA or https://www.bmj.com/content/322/7280/226.1) advise against dichotomising results based on arbitrary p-values. While clearly many researchers do continue to use the language of "statistical significance" despite such recommendations, we would at least suggest the authors consider avoiding what many consider to be outmoded practice.

10. We were not able to open either the pdf or MS Word versions of the Reporting Standards Document "CONSORT Checklist".

[Editors’ note: further revisions were suggested prior to acceptance, as described below.]

Thank you for resubmitting your work entitled "Effectiveness of rapid SARS-CoV-2 genome sequencing in supporting infection control for hospital-onset COVID-19 infection: multicenter, prospective study" for further consideration by *eLife*. Your revised article has been evaluated by Jos van der Meer (Senior Editor) and a Reviewing Editor.

The manuscript has been improved but there are some remaining issues that need to be addressed, as outlined below:

In an ideal world, it would be nice if there was more framing of the work in the abstract, intro, and discussion to highlight that the study took place during an earlier phase of the pandemic when there was less insight into viral transmission, gaps in public health teaching about how best to prevent nosocomial transmission of SARS-CoV-2, and dire shortages of PPE, personnel, beds, and lab capacity. These are all important caveats that luckily are less acute now but strongly impact the interpretation of the study. I wish these caveats were pronounced more prominently in the paper in order to defend against the overly simplistic conclusion that WGS does not aid infection control of SARS-CoV-2.

---

## [Author Response]

Reviewer #1 (Recommendations for the authors):Interpretational questions, commentsThe turnaround times achieved are, definitely for the "rapid" part, much longer than was targeted, with overall 9,3% reported in time. This implies that even in the settings tested (which I consider "as good as it gets") these targets were too optimistic. Yet, there is a clear difference in reporting time between the 2 strategies, and that difference, as achieved, was not associated with a beneficial outcome on any of the measured outcome parameters. In fact, the proportion of HOCI in which an unexpected linkage was detected based upon WGS was low in both study phases. And the changes in IPC that were realized were infrequent (7.4-7.8%), and associated with short turnaround times. My interpretation would be that reporting of WGS was still too slow to act as a true intervention, and, therefore, no effects could be demonstrated.

We have added a further explicit acknowledgments that the target turnaround times were not met in the Results:

“The median turnaround time from diagnostic sampling for reports returned was 5 days in the rapid phase and 13 days in the longer-turnaround phase, substantially longer than the targets of 48 hours and 5-10 days, respectively. A detailed breakdown of reporting turnaround times is reported separately[15].”,

and in the Discussion

“The study sites varied in their ability to process sequence and meta-data and generate and distribute reports in a timely manner (Figure 1), and the targeted turnaround times for reporting were not achieved at any of the sites for the majority of HOCIs in either the ‘rapid’ or ‘longer turnaround’ phases.”

We have also added a comment that improved turnaround times might lead to a greater impact on IPC practice:

“It may therefore be more achievable to develop effective systems for rapid viral WGS and feedback for endemic respiratory viruses at lower and more consistent levels, and more timely reporting of results might be associated with greater impact on IPC actions.”

Nevertheless, it was stated that "Extensive qualitative analyses[15] found high levels of acceptability for the SRT sequencing reports, which supported decision-making about IPC activity (e.g. stand down some IPC actions or continue as planned)."

With regards to the high acceptability of the intervention for IPC teams, we have noted in the existing text describing the qualitative analyses that “The SRT did provide new and valued insights into transmission events, outbreaks and wider hospital functioning but mainly acted to offer confirmation and reassurance to IPC teams”. As such, high acceptability of the intervention does not conflict with the high proportion of cases in which reports were considered supportive of IPC actions already taken.

Please note that we have also added a reference to a paper specifically describing qualitative analysis of the acceptability of whole genome sequencing for infection control associated with this study, which is now available on medRxiv:

Flowers P, McLeod J, Mapp F, Stirrup O, Blackstone J, Snell L, et al. How acceptable is rapid whole genome sequencing for infectious disease management in hospitals? Perspectives of those involved in managing nosocomial SARS-CoV-2. medRxiv 2022:2022.2006.2015.22276423.

This work is separate to the qualitative process evaluation for the present study (which is not yet available as a preprint):

Mapp F, Flowers P, Blackstone J, Stirrup O, Copas A, Breuer J. Understanding the effectiveness of rapid SARS-CoV-2 genome sequencing in supporting infection control teams through qualitative process evaluation: the COG-UK hospital-onset COVID-19 infection study. Submitted 2022.

I think that the authors are overly optimistic about the impact of WGS reporting in this study in the discussion:Line 295: "Our study provides a substantial body of evidence regarding the introduction of viral WGS into hospital functioning, routine IPC practice, its potential impact on outbreak management and the challenges that need to be overcome to achieve implementation across the UK." I don't understand what is meant with "that the study provides evidence for its potential impact", as statistically significant impact is not demonstrated for any of the study outcomes.

As the sequencing intervention was qualitatively reported to be useful for the majority of cases in which SRT reports were returned and as there was an impact on IPC actions in some HOCI cases, we feel that it is reasonable to discuss a ‘potential impact on outbreak management’ despite the fact that we did not demonstrate a reduction in the incidence of HAIs during the study intervention phases (particularly as we know that the capacity of IPC teams to respond was breached during peaks in SARS-CoV-2 cases). However, to moderate the tone of this statement, we have edited ‘impact on’ to ‘usage for’ here.

Line 346: "Our study provides the first evidence that with faster turnaround times, viral sequences can inform ongoing IPC actions in managing nosocomial infections; results returned within {less than or equal to}5 days from sampling to result changed the actions of IPC teams in around 20% of cases." On an intention-to-treat basis there was no significant effect on the occurrence of HOCI, and it doesn't become clear from Tables S2 and S3 that IPC practices were changed

Tables S2 and S3 only include pre-specified options that were listed on reporting forms developed during the summer of 2020. However, we have now clarified in the Methods that

“There was considered to be an impact on IPC actions if this was recorded for any of a number of pre-defined outcomes, or if it was stated that the report had effected any change to IPC practice on that ward or elsewhere within the hospital”,

and added in the Results ‘among the options included within study reporting forms’. As noted in the Methods, a full description of outcome variable coding is provided in the Supplementary Appendix.

It is not clear from these tables if – in instances where measures were taken – more measures were taken simultaneously. That could be made more clear. For instance in Table S3 for Rapid and support: were measures taken in 5 of 70 outbreaks or in 30 of 70 (if you sum up all individual measures).

The proportion of HOCI cases with any impact on IPC actions recorded is summarised in Tables 2 (for all cases) and S1 (for the first case within each defined outbreak). This has now been flagged in the footnotes for Tables S2 and S3.

The research question remains whether even shorter turnaround times for more patients with HOCI could have an impact on transmissions and – ultimately – patient outcome.

We agree with this point, and have added the following sentence at the end of the Discussion

“It remains to be demonstrated that viral sequencing can have a direct impact on clinical outcomes such as the incidence of HAIs, and further prospective studies with refined implementation of similar interventions are required to address this”.

I would think that WGS could only have such an impact if IPC measures would be different for different genotypes or would be different if transmission was demonstrated. The authors could better explain why and under what circumstances they expect that WGS reporting could impact patient care.

We have added the following to the Discussion, particularly with reference to development of an improved reporting system with more detailed patient movement data and transmission network analysis:

“Implementation of an improved tool with these features might help to better identify routes of transmission within a hospital that could be interrupted, e.g. through changes to the management of ward transfers for patients, isolation policies or identification of areas within the hospital linked to high risk of transmission.”

We also now note in the Discussion that

“In the planning of an equivalent study now, there would be a greater focus on adjustments to ventilation, air filtration and respirator usage.”

Reviewer #2 (Recommendations for the authors):1. The study appears to have excluded cases where linkage between cases was established on epidemiological grounds before sequencing occurred. This both makes it more difficult to see an effect of sequencing and begs the question about the intensity and comparability of IPC programs across the study hospitals. Hospitals with aggressive and capable IPC teams may have very good insight into nosocomial case sources and factors by virtue of their epidemiologic evaluations and thus have less room to benefit from sequencing.

We did not exclude any HOCI cases on the basis that linkage had already been observed prior to sequencing. However, the Reviewer is correct that for the second primary outcome “identification of linkage to individuals within an outbreak of SARS-CoV-2 nosocomial transmission using sequencing … that was not identified by pre-sequencing IPC evaluation”, there may have been ‘less room for improvement’ with more thorough initial IPC investigation. We have clarified the evaluation of this outcome with the following new sentence in the Methods

‘The second outcome used all observed HOCI cases as the denominator, and so represented the proportion of cases in which sequencing provided information regarding potential transmission routes where none had been previously uncovered.’

2. The investigators report that nosocomial infection was not suspected in ~7% of HOCI cases. Please contextualize by stating what % of HOCI cases were suspected to be nosocomial.

All included HOCI cases were suspected by IPC teams to be nosocomial infections, although there will have been some uncertainty for the 899 (41.4%) indeterminate cases diagnosed 3-7 days after admission. We have clarified interpretation of the ~7% values in the Results:

‘Nosocomial linkage to other individual cases, where initial IPC investigation had not correctly identified any such linkage, was identified in 6.7% and 6.8% of all HOCI cases in the rapid and longer-turnaround phases, respectively’.

3. It would be helpful to describe in the main body of the manuscript the kinds of actions that were expected of IPC programs and how they might decrease hospital-acquired infections.

We have added the following to the Results ‘Guidance regarding IPC actions was not specified as part of this study. Sites were expected to follow current national guidelines, which evolved throughout the course of the pandemic.’ However, we have now mentioned within the main text of the Methods some of the pre-defined changes to IPC actions recorded on study forms:

‘There was considered to be an impact on IPC actions if this was recorded for any of a number of pre-defined outcomes (e.g. enhanced cleaning, visitor and staffing restrictions, provision of personal protective equipment), or if it was stated that the report had effected any change to IPC practice on that ward or elsewhere within the hospital.’

As noted in response to the public comments above, we have also added the following in the Discussion:

“Our qualitative analyses also found that the capacity of sites … was breached by the volume of HOCI cases in combination with the finite personnel resources and limited physical space for isolation that was available” and “Planning this study … during the early stages of a novel viral pandemic was challenging… In the planning of an equivalent study now.. It would also be possible to be more prescriptive and standardised regarding the recommended changes to IPC practice in response to sequencing findings”.

4. Please clarify the criteria for higher performer sites. On the face of it, SRT in ~40% of cases and median time from diagnostic sample to SRT of up to 8 days doesn't seem like high performance relative to the study goals.

We have added the following to the Results:

“The criteria for this analysis were decided after data collection but prior to data analysis, as per the statistical analysis plan. However, we acknowledge that the ‘higher performing sites’ did not meet the target turnaround time for reporting in the rapid phase; criteria were therefore set to split the sites into upper and lower 50% based on level of implementation”.

As stated in the Results, the ‘higher performing sites’ “…returned ≥40% of SRTs within a median time from diagnostic sample of ≤8 days within their rapid phase”. These criteria were used to select the sites in the bottom right corner of the ‘Rapid phase’ plot in Figure 1.

Were there other aspects to these sites that differentiated them besides sequencing speed? Did they have larger or more experienced or more aggressive infection control teams just as they appear to have more effective sequencing teams?

Yes, there were differences between the sites in their staffing and organisation of IPC teams, but these were not formally investigated. Some discussion regarding how this impacted on implementation of the HOCI intervention will be included in the full qualitative process evaluation, for example: routinisation of discussion of sequencing reports, dynamic leadership, effective and rapid sharing of sequencing reports. We have now switched to analysis of a fully anonymised data without site names for data governance reasons and are unable to link the study data back to individual sites.

5. Was there a standardized protocol for how IPC teams were to respond to sequencing data? Without this it's possible that opportunities existed to reduce transmission based on sequencing reports that were not taken.

Please see response to comment #3 above. We agree that opportunities to reduce transmission will have been missed due to imperfect understanding of the virus and optimal strategies, limited staff and PPE and constraints associated with aging buildings and chronic underinvestment in the health service.

In the planning of the study, there was considerable debate about guidance for IPC teams. Given the heterogeneity of the sites and their resources it was felt that it was best left for sites to generate local solutions. The sites were not in a position to routinely follow systematic guidance for IPC reactions to sequencing data in winter 2020/2021.

6. Did study sequencing and SRTs incorporate healthcare worker test results? If so, please describe testing requirements for healthcare workers. Did they get serial testing or only in response to symptoms or exposures? If cluster detection did not include healthcare worker test results then this could lead to substantial underdetection of nosocomial events as well as obscure to IPCs the source of nosocomial cases and thus how best to contain them.

We have added the following text to the Methods:

‘Sequencing data from healthcare workers (HCW) could be utilised in the SRT system, and this was implemented by 8/14 sites. Whether this was done depended on availability of HCW samples for each lab as staff testing was generally managed separately to patient testing. HCW testing protocols followed national guidelines’.

We have also added the following to the Discussion:

‘HCW sequencing data could not be incorporated at all sites dues to logistical and data management and access constraints’.

7. Please comment on the median incubation period for hospital-onset cases. With α and subsequent variants, incubation periods have become quite short thus progressively increasing the importance of speed in sequencing to have a potential impact on infections.

We had mentioned the potentially lower incubation time of the Α variant in the Discussion, but we have now also added the following:

‘Variants with shorter incubation times would lead to a greater importance for the rapidity of feedback in informing adjustments to IPC actions’.

8. In our hospital experience, the highest value from sequencing has not been control of live outbreaks but rather insights gained regarding the robustness of global infection control policies (is a surgical mask protective, do AGPs increase risk, what is the risk of transmission between people in shared rooms, etc.). Hospitals that choose to act on these insights will make structural changes to their IPC programs (e.g. switch from surgical mask to respirators, change testing practices around AGPs, decrease use of shared rooms, increase ventilation within rooms, etc.) that ideally will not just be cluster-specific but lasting changes. This means that the incremental value of sequencing may decrease over time to the extent that one puts these successive changes into place. But sequencing itself was invaluable in providing the insights that led to these changes. In addition, these insights and policy changes are arguably less sensitive to turnaround time – the insights themselves are key but the responses transcend any one cluster and thus are less time sensitive.

Thank you for flagging this. We agree with the points raised, and have added the following to the Discussion:

‘As well as acute changes to IPC actions, there is the potential for routine pathogen sequencing to allow prospective IPC practice and policies to be refined. This could enable a longer-term reduction in the incidence of nosocomial infection at any given site, and such effects would be less dependent on turnaround time of sequencing in any given case. However, the capacity of sites to make such informed adjustments to IPC practice were limited during peaks in incidence of SARS-CoV-2 over the timescale of the present study’.

Reviewer #3 (Recommendations for the authors):1. The potential effect of the intervention depends on infection prevention measures that would have been taken in the absence of sequencing, and how hospitals' infection prevention and control teams respond to the sequencing results.

We have added to the Methods ‘Guidance regarding IPC actions was not specified as part of this study. Sites were expected to follow current national guidelines, which evolved throughout the course of the pandemic’, and also relevant text to the Results:

“Planning this study … during the early stages of a novel viral pandemic was challenging… In the planning of an equivalent study now.. It would also be possible to be more prescriptive and standardised regarding the recommended changes to IPC practice in response to sequencing findings”.

We do not feel that it would have been possible to be prescriptive regarding IPC actions in the summer of 2020, particularly given differences between sites in IPC organisation and available resources.

There is therefore considerable potential for variability in the effect of the intervention between hospitals. From the methods it is not clear how analyses accounted for the effect of differences between hospitals.

The analysis models included site-phase random effect terms, which allow for variability in the effect of the intervention. However, we were aiming to find evidence of an overall reduction in the incidence of nosocomial infections across the sites. Analyses did not directly adjust for differences between hospitals in IPC actions.

More generally, there is a lack of transparency as to precisely what the modelling assumptions were. It is important to either specify the models used nambiguously using appropriate notation (including details on link functions and all random effects) in the methods or to provide the full STATA code used for the analysis (ideally, of course, both of these things should be done).

We feel that the original text of the manuscript does give an appropriate degree of detail regarding the models used for statistical analysis. However, we have added further clarification to the Appendices as requested within a new subsection

‘Further details for incidence model specification’ which includes indicative Stata code. We have also added to the Statistical Analysis subsection of the main text that such models ‘..in this context correspond to Poisson regression with an additional overdispersion parameter’.

It also important to specify which analyses were pre-specified in a statistical analysis plan and which were unplanned.

For the most part the analyses presented were prespecified in the SAP. However, we have clarified in the text that the results in Figure 2 go beyond the evaluation specified in the SAP, and we have ensured that the ‘Changes with respect to the statistical analysis plan (SAP)’ subsection in the Appendix is fully comprehensive.

2. What the authors refer to as the "per protocol" sensitivity analysis in fact seems to be an unplanned sub-group analysis. Given that none of the trusts completely adhered to the protocol referring to this as a "per protocol" analysis seems misleading.

We have added the following clarification in the ‘Changes with respect to the statistical analysis plan’ section of the Appendix:

“It was stated in the SAP that “We will conduct sensitivity analyses excluding study sites and/or periods with suboptimal implementation of the trial intervention, both in terms of overall population sequencing coverage for HOCIs and the turnaround time for sequence reports being returned to IPC teams. The exact criteria for this will be decided amongst the study team before any analysis has been conducted”. This forms the basis for the ‘per protocol’ analyses presented. It was not possible to prespecify the exact criteria. After data collection for the study had been completed it became clear that none of the sites had met target turnaround times for sequence reporting in the intervention phases, and so it was decided to set criteria to select the 50% ‘higher performing’ sites.”

We feel that ‘per protocol’ is a useful concise term to describe this sensitivity analysis, but have added inverted commas throughout and have now noted in the Results that:

“The criteria for this analysis were decided after data collection but prior to data analysis, as per the statistical analysis plan (SAP). However, we acknowledge that the ‘higher performing sites’ did not meet the target turnaround time for reporting in the rapid phase; criteria were therefore set to split the sites into upper and lower 50% based on level of implementation.”

3. It is also important to see the difference between hospitals displayed in the results. Taking Figure 2 as an example, as the results are currently presented it is impossible to tell whether trends with increased time to result represent trends within each hospital, or a difference between those hospitals that tend to produce rapid results compared to those that tend to produce less rapid results. It should be a simple matter to produce a graph showing unaggregated results from each hospital together with the fitted model (and prediction intervals).

We have added these plots as requested and added a comment in the Results:

‘However, we note that many of the HOCI cases with SRT returned within 5 days were from a single study site, and some sites did not seem to have clearly differentiated ‘useful’ SRT reports when completing data collection (Figures S2 and S3)’.

4. As the authors discuss, this trial took place during a period of significant variation in covid-19 prevalence, both in the community and among hospital patients. They report having adjusted their results for this. It would be useful to see both unadjusted and adjusted results to show the impact of this adjustment.

We do not feel that unadjusted IRR estimates are meaningful given the lack of randomisation, and the clear dependence of the outcomes on the adjustment variables. However, we have now added these values for completeness in Appendix 1—table 5 and noted this additional deviation from the SAP.

5. One hospital did not participate in the longer turnaround arm as this was considered inferior to their standard practice. Please describe current levels of whole genome use in these hospitals so that it is explicit how the intervention differs from the baseline conditions.

We have added a clarification to the Methods: ‘, comprising outbreak sequencing with weekly meetings to discuss phylogenetic analyses; they nonetheless completed the baseline phase of the study without use of the SRT or automated feedback to IPC teams on all HOCI cases’.

6. Please re-check the odds ratio on line 222. The rapid sequencing has a greater proportion of finding it helpful so the odds ratio should be greater than 1. If this is correct due to adjustment, please report this as adjusted OR and add details of adjustment.

Thank you for flagging this, we have added a clarification: ‘although this association was reversed on analysis within the multi-level mode specified,’ and have also now specified in the Methods that mixed effect logistic regression models were also used for ‘…the ‘usefulness’ of SRT reports’.

7. It is important that the discussion starts with a clear statement that for the two primary outcomes no benefit of either intervention was shown and that this was true both in the main analysis and in the sensitivity analysis

We have added a more direct statement at the opening of the Discussion as requested:

‘We did not demonstrate a direct impact of sequencing on the primary outcome of the incidence of HAIs, either on full analysis or when restricted to the higher performing sites, and the overall proportion of cases with nosocomial transmission linkage identified using sequencing that had been missed by IPC investigation was <10% in the intervention phases’.

8. Line 274 "Among sites with the most effective implementation of the sequencing intervention we showed that feedback within 5 days of diagnosis allowed for maximal impact on IPC actions". This appears to be an unplanned subgroup analysis of a secondary outcome, and as such needs to come with suitable caveats.

As noted above, this exploratory investigation was not entirely unplanned. However, we appreciate this point and have added ‘…post hoc exploratory investigation…’ here.

9. Line 317 " may reflect a shorter incubation time for the Α variant compared to earlier variants" While there has been speculation about shorter incubation times for the Α variant, we are not aware of strong evidence that supports this, and recent analysis suggests similar generation time for the Α variant and earlier variants (see, for example, https://doi.org/10.1101/2021.12.08.21267454 which has now been accepted for publication in eLife).

Thank you for flagging this work, we have added this as a reference at the relevant point in the Discussion: ‘(although this remains uncertain^[31]^)’.

10. Line 345. "Our study provides the first evidence that with faster turnaroundtimes, viral sequences can inform ongoing IPC actions in managing nosocomial infections; " While generally the paper strikes an appropriate note of cautious optimism for the approach despite the negative results in this trial, and it is understandable that the authors might want to emphasize the more positive aspects of the results when considering implications for sequencing to inform IPC measures, there is a danger that this statement (and the whole paragraph) is interpreted as attempting to put a positive spin on an essentially negative result. Thus, while it is reasonable to mention that in a minority of cases IPC actions were informed by viral sequencing data, it is equally important to emphasize that there was no evidence those measures had a positive impact on patient outcomes (it is surely equally possible that IPC actions informed by sequencing had a negative impact). Furthermore, it needs to be emphasized that in the main (planned) analysis there was no difference in the number of IPC actions informed by the sequencing results in rapid and longer-turnaround arms. It would be helpful to add here an explicit acknowledgement of the uncertain benefits and resource implications in this paragraph along with a discussion of how we might robustly evaluate the effectiveness (and cost-effectiveness) of WGS methods for IPC in the future, and in particular, emphasising what we learn from this trial.

We have added some caveats here to the text as requested:

‘We did not demonstrate an effect of our sequencing intervention on our primary outcome of the incidence of HAIs, and there were challenges in the implementation of the intervention. However, our study provides the first prospective evidence that with faster turnaround times, viral sequences can inform ongoing IPC actions in managing nosocomial infections; on post hoc exploratory analysis, results returned within ≤5 days from sampling to result changed the actions of IPC teams in around 20% of cases’.

Given these reminders of the limitations of our results, we feel that this ‘positive’ paragraph is reasonable in the context of the Discussion as a whole, particularly with the changes to the first and last paragraphs in this revised version.

11. Lines 596-602 Two planned secondary outcomes are not reported because they were either not thought to be useful or were insufficiently complete. While it seems reasonable to not conduct formal analysis for these reasons, surely these outcomes should still be reported somewhere given they were specified in the protocol and data does seem to have been obtained for them. Clearly, if researchers are permitted to selectively choose which of the planned outcomes to report there is large potential for bias.

The available descriptive data for these outcomes are summarised in Tables 2 and 3. We have also clarified at the relevant point in the Appendices that the decision to drop formal analysis was made prior to any statistical analysis being conducted, and that data collection for HCW absence ceased during the conduct of the study, reducing the potential for bias through selective reporting. We feel that dropping these outcomes is reasonable given the complexity and pandemic context of the study, and that we have been transparent in our reporting.

Additional comments1. line 528-9 "Site A observed 0 HOCI cases " – presumably this is just for longer-turnaround phase, as A appears on the graph in the rapid phase.

Yes, this sentence is regarding the longer-turnaround phase ‘Site N did not have a longer-turnaround phase, Site A observed 0 HOCI cases …in this phase’.

2. Figures S3 and S4: solid and dashed lines are not defined. Also, shading might be clearer than the dashed lines.

We have added the following to the captions for each of these figures: ‘The associations for each covariable indicated by model parameter point estimates are shown as solid lines, with 95%CIs shown as dashed lines’. The use of dashed or dotted lines is conventional for plotting 95%CIs.

3. In Table 3, a number of outcomes are reported as 0.0 per 100 patients. It would be more helpful to display these in a unit that allows easy display of at least one significant figure, e.g. per 10 000.

We have changed the units for outbreak events to ‘weekly incidence per 1000 inpatients’ to improve the display.

4. Some figures use the colours red and green as the only way to distinguish between baseline and rapid periods. The additional distinction might help readers who are unable to distinguish between these colours.

Thank you for this comment, we have switched these figures to a colour-blind friendly palette.

5. The results have a subsection titled health economics. This is limited to laboratory costs, and has no reference to changes in outcomes or costs or savings associated with changing practice on the wards. We suggest changing the title to costs of tests.

Thank you for your comment. The scope of the economic evaluation was to evaluate the economic effects of SARS-CoV-2 genome sequencing in supporting infection control teams and not the health benefits of the intervention. Therefore, the outcome of the intervention would be rather the benefit of rapid/slow return of the sequencing report expressed as potential reduction in resource utilisation and costs. A paper presenting the methodology and findings is under preparation. Here, we presented only the cost of the intervention. We will remove the “Health economic findings” heading and re-name it “Cost of SARS-CoV-2 genome sequencing”.

6. Lab 9 appears to have had larger batch sizes and a significant cost saving associated with rapid testing. Is this due to the difference in prevalence at the different times they were following each protocol, or is there a difference in practice that could be applied more widely?

Thank you for your comment. The costing analysis was based entirely on information provided by laboratories. Through the various phases of the study, the situation changed drastically, for example depending upon sample numbers and the required turnaround the pathway adopted was adapted by each laboratory. Generally, costs can vary significantly depending on run size, kits/reagents, staff, with small batches working out more expensive per sample. Lab 9 did use different sequencing platforms in intervention phases: Illumina MiSeq sequencing with 96 samples per run in rapid phase when there was a higher throughput; for lower throughput, as the sample numbers declined, they switched to Nanopore ONP in slower turnaround phase.

7. It is interesting to note how much more frequently teams reported finding the report useful compared to how often they reported using it to change practice. Was any information collected on this? Were they reporting reassurance from the results or were they useful in some other way?

Yes, we feel that reassurance regarding previous conclusions and actions is likely to main explanation here. This is consistent with SRT reports being reported to be consistent with prior IPC actions in >90% of HOCI cases (Appendix 1—table 2). ‘Reassurance’ is also mentioned in our brief summary of the qualitative analysis in the Results, with the full process evaluation yet to be published.

8. Please define IRR on first use in the text and abstract.

We have now done this, thank you for flagging.

9. Abstract and line 200 "We did not detect a statistically significant change".Recommendations in the statistical literature (e.g. consensus statement from the ASA or https://www.bmj.com/content/322/7280/226.1) advise against dichotomising results based on arbitrary p-values. While clearly many researchers do continue to use the language of "statistical significance" despite such recommendations, we would at least suggest the authors consider avoiding what many consider to be outmoded practice.

Thank you for your comment. We appreciate the sentiment, but feel that the use of this phrase is reasonable in the context of the primary outcome of a prospective study.

10. We were not able to open either the pdf or MS Word versions of the Reporting Standards Document "CONSORT Checklist".

This would seem to be an issue regarding the *eLife* file system.

[Editors’ note: further revisions were suggested prior to acceptance, as described below.]

In an ideal world, it would be nice if there was more framing of the work in the abstract, intro, and discussion to highlight that the study took place during an earlier phase of the pandemic when there was less insight into viral transmission, gaps in public health teaching about how best to prevent nosocomial transmission of SARS-CoV-2, and dire shortages of PPE, personnel, beds, and lab capacity. These are all important caveats that luckily are less acute now but strongly impact the interpretation of the study. I wish these caveats were pronounced more prominently in the paper in order to defend against the overly simplistic conclusion that WGS does not aid infection control of SARS-CoV-2.

Thank you for these further comments. We have made some adjustments throughout the manuscript in response to the concerns raised.

We have added the following to the Abstract of the manuscript, in order to flag the context of the study as requested:

“A total of 2170 HOCI cases were recorded from October 2020-April 2021, corresponding to a period of extreme strain on the health service, with sequence reports returned for…”

And:

“Capacity to respond effectively to insights from sequencing was breached in most sites by the volume of cases and limited resources.”

These additions mean that the Abstract is now a little over the standard 250 word count limit recommended by *eLife*. We feel that these additions are justified, as suggested by your comments, and that there is not any other information that could be removed from the Abstract. We hope that the Editors agree that the Abstract word limit may be extended a little in this case.

We have added the following paragraph at the end of the Introduction:

“When this study was planned in the summer of 2020 there was imperfect knowledge regarding the dominant mode of transmission of SARS-CoV-2[10] and it was not possible to predict the future course of the pandemic. In conducting this study, substantial difficulties were encountered in implementing the intervention and in responding effectively to any insights generated. As such, this report serves as a record of the challenge of conducting research within a pandemic as well as being a conventional study summary report.”

With reference:

10. Greenhalgh T, Jimenez JL, Prather KA, Tufekci Z, Fisman D, Schooley R. Ten scientific reasons in support of airborne transmission of SARS-CoV-2. *The Lancet* 2021; 397(10285):1603-1605.

In the Results section, we have added:

“Critically, given the context of the study within the pandemic timeline, the capacity to generate and respond to these insights effectively on a case-by-case basis was breached in most sites by the volume of HOCIs”

We feel that the Discussion did already cover a number of the issues raised by the Editors. However, in order to highlight these points more clearly to the reader, we have added the following paragraph near the start of the Discussion:

“The study was undertaken during a period of extreme strain on the NHS, with hospitals described as being “in the eye of a covid-19 storm”[19]. Sites reported that they lacked the additional resources, in terms of staff and bed space, needed to respond effectively to insights generated by sequencing. Furthermore, if the study were repeated now then IPC teams would have more evidence-backed tools at their disposal, such as increasing respirator usage. As such, we do not believe that the null result for the impact on incidence of nosocomial transmission should be taken as strong evidence for a general lack of effectiveness of viral WGS for IPC.”

With reference:

19. Oliver D. David Oliver: Could we do better on hospital acquired covid-19 in a future wave? *BMJ* 2021; 372:n70.

We have also made edits and added small sections of additional text at a few points elsewhere in the Discussion:

“…the challenges faced in implementing the intervention reflected the context and barriers in winter 2020–2021 in the UK”

“It would also be possible to be more prescriptive and standardised regarding the recommended changes to IPC practice in response to sequencing findings, with the potential that our improved knowledge and available tools might facilitate a measurable impact on the incidence of nosocomial transmission.”

“Our study nonetheless provides valuable evidence regarding the implementation and utility of this technology for IPC, and potentially it will have a greater positive impact on IPC practice outside of the burdens and resource constraints imposed by a pandemic”

We have added a couple of direct quotes from IPC staff, taken from our qualitative analyses, in order to further communicate the context of this study:

“Our qualitative analyses also found that the capacity of sites to react to information generated by the sequencing intervention was breached by the volume of HOCI and admitted COVID-19 cases (‘we’ve been basically deluged’ IPC staff) in combination with the finite personnel resources and limited physical space for isolation that was available (‘The trouble is when you have so many wards going down and such a high prevalence of COVID, your actions are kind of the same regardless’ IPC staff).”

We have also added a few further relevant references to the Discussion:

“In the planning of an equivalent study now, there would be a greater focus on adjustments to ventilation[32], air filtration[33] and respirator[34] usage.”

32. Allen JG, Ibrahim AM. Indoor Air Changes and Potential Implications for SARS-CoV-2 Transmission. *JAMA* 2021; 325(20):2112-2113.

33. Conway Morris A, Sharrocks K, Bousfield R, Kermack L, Maes M, Higginson E, et al. The Removal of Airborne Severe Acute Respiratory Syndrome Coronavirus 2 (SARS-CoV-2) and Other Microbial Bioaerosols by Air Filtration on Coronavirus Disease 2019 (COVID-19) Surge Units. *Clinical Infectious Diseases* 2021:ciab933.

34. Ferris M, Ferris R, Workman C, O'Connor E, Enoch DA, Goldesgeyme E, et al. Efficacy of FFP3 respirators for prevention of SARS-CoV-2 infection in healthcare workers. *eLife* 2021; 10:e71131.